# How Do Fair Decisions Fare in Long-term Qualification?

**Xueru Zhang**[1,∗]      **Ruibo Tu**[2,∗]      **Yang Liu**[3]      **Mingyan Liu**[1]
**Hedvig Kjellström**[2]      **Kun Zhang**[4]      **Cheng Zhang**[5]

[1]University of Michigan, {`xueru,mingyan`}`@umich.edu`
[2]KTH Royal Institute of Technology, {`ruibo,hedvig`}`@kth.se`
[3]University of California, Santa Cruz, `yangliu@ucsc.edu`
[4]Carnegie Mellon University, `kunz1@cmu.edu`
[5]Microsoft Research, `Cheng.Zhang@microsoft.com`

## Abstract

Although many fairness criteria have been proposed for decision making, their long-term impact on the well-being of a population remains unclear. In this work, we study the dynamics of population qualification and algorithmic decisions under a partially observed Markov decision problem setting. By characterizing the equilibrium of such dynamics, we analyze the long-term impact of static fairness constraints on the equality and improvement of group well-being. Our results show that static fairness constraints can either promote equality or exacerbate disparity depending on the driving factor of qualification transitions and the effect of sensitive attributes on feature distributions. We also consider possible interventions that can effectively improve group qualification or promote equality of group qualification. Our theoretical results and experiments on static real-world datasets with simulated dynamics show that our framework can be used to facilitate social science studies.

## 1 Introduction

Automated decision making systems trained with real-world data can have inherent bias and exhibit discrimination against disadvantaged groups. One common approach to alleviating the issue is to impose fairness constraints on the decision such that certain statistical measures (e.g., true positive rate, positive classification rate, etc.) across multiple groups are (approximately) equalized. However, their effectiveness has been studied mostly in a static framework, where only the immediate impact of the constraint is assessed but not its long-term consequences. Recent studies have shown that imposing static fairness criteria intended to protect disadvantaged groups can actually lead to pernicious long-term effects [33, 47]. These long-term effects are heavily shaped by the interplay between algorithmic decisions and individuals' reactions [34]: algorithmic decisions lead to changes in the underlying feature distribution, which then feeds back into the decision making process. Understanding how this type of coupled dynamics evolve is a major challenge [10].

Toward this end, we consider a discrete-time sequential decision process applied to a certain population, where responses to the decisions made at each time step are manifested in changes in the features of the population in the next time step. Our goal is to understand how (static) fairness criteria in this type of decision making affect the evolution of group well-being and characterize any equilibrium state the system may converge to. In particular, we will focus on *myopic* policies that maximize the immediate utility under static fairness constraints, and examine their impact on different groups in the long run.

---

∗Equal contribution

More specifically, we seek to study the dynamics of group qualification rates [30, 34, 37, 44] and evaluate the long-term impact of various static fairness constraints imposed on decision making. We examine whether these static fairness constraints mitigate or worsen the qualification disparity in the long-run. Our work can be applied to a variety of applications such as recruitment and bank lending. In these applications, an *institute* observes *individuals'* features (e.g., credit scores), and makes *myopic decisions* (e.g., issue loans) by assessing such features against some variables of interest (e.g., ability to repay) which are unknown and unobservable to the institute when making decisions. Individuals respond to the decisions by investing in effort to either improve or maintain their qualification in the next time step. These actions collectively change the qualification rate of the population. In summary, our main contributions are:

1. *We analyze the equilibrium of qualification rates in different groups under a general class of fairness constraints (Section 4).* We use a Partially Observed Markov Decision Process (POMDP) framework to model the sequential decision making in different scenarios (Section 3). Using this model, we show that under our formulation optimal policies are of the threshold type and provide a way to compute the threshold. We then prove the existence of an equilibrium (in terms of long-term qualification rates) using threshold policies and provide sufficient conditions for a unique equilibrium.
2. *We analyze the impact of fairness constraints on the disparity of qualification rates* when the equilibrium is unique (Section 5). Our findings suggest that the same fairness constraint can have opposite impacts on the equilibrium depending on the underlying problem scenario.
3. *We explore alternative interventions that can be effective in improving qualification rates at the equilibrium and promoting equality across different groups (Section 6).*
4. *We examine our theory on synthetic Gaussian datasets and two real-world scenarios (Section 7).* Our experiments show that our framework can help examine findings cross domains and support real-life policy making.

## 2  Related Work

Among existing works on fairness in sequential decision making problems [45], many assume that the population's feature distribution neither changes over time nor is it affected by decisions; examples include studies on handling bias in online learning [6, 11–13, 16, 20, 28, 31] and bandits problems [4, 8, 26, 27, 32, 35, 39, 43]. The goal of most of these work is to design algorithms that can learn near-optimal policy quickly from the sequentially arrived data and the partially observed information, and understand the impact of imposing fairness intervention on the learned policy (e.g., total utility, learning rate, sample complexity, etc.)

However, recent studies [2, 7, 15] have shown that there exists a complex interplay between algorithmic decisions and individuals, e.g., user participation dynamics [19, 46, 47], strategic reasoning in a game [23, 30], etc., such that decision making directly leads to changes in the underlying feature distribution, which then feeds back into the decision making process. Many studies thus aim at understanding the impacts of imposing fairness constraints when decisions affect underlying feature distribution. For example, [33, 21, 29, 30] construct two-stage models where only the one-step impacts of fairness intervention on the underlying population are examined but not the long-term impacts in a sequential framework; [24, 38] focus on the fairness in reinforcement learning, of which the goal is to learn a long-run optimal policy that maximizes the cumulative rewards subject to certain fairness constraint; [19, 47] construct a user participation dynamics model where individuals respond to perceived decisions by leaving the system uniformly at random. The goal is to understand the impact of various fairness interventions on group representation.

Our work is most relevant to [23, 34, 37, 44], which study the long-term impacts of decisions on the groups' qualification states with different dynamics. In [23, 34], strategic individuals are assumed to be able to observe the current policy, based on which they can manipulate the qualification states strategically to receive better decisions. However, there is a lack of study on the influence of the sensitive attribute on dynamics and impact of fairness constraints. Besides, in many cases, the qualification states are affected by both the policy and the qualifications at the previous time step, which is considered in [37, 44]. However, they assume that the decision maker have access to qualification states and the dynamics of the qualification rates is the same in different groups, i.e.,the equally qualified people from different groups after perceiving the same decision will have the same future qualification state. In fact, the qualification states are unobservable in most cases, and the dynamics can vary across different groups. If considering such difference, the dynamics can be much more complicated such that the social equality can not be attained easily as concluded in [37, 44].

# 3 Problem Formulation

**Partially Observed Markov Decision Process (POMDP).**
Consider two groups $\mathcal{G}_a$ and $\mathcal{G}_b$ distinguished by a sensitive *attribute* $S = s \in \{a, b\}$ (e.g., gender), with fractions $p_s := \mathbb{P}(S = s)$ of the population. At time $t$, an *individual* with attribute $s$ has feature[2] $X_t = x \in \mathbb{R}$ determined by a hidden *qualification* state $Y_t = y \in \{0, 1\}$, and both are time-varying. We adopt a natural assumption that an individual's attribute and current features constitute sufficient statistics, so that conditioned on these, the decision is independent of past features and decisions. This allows an *institute* (decision maker) to adopt a Markov policy: it makes decisions $D_t = d \in \{0, 1\}$ (reject or accept) using a policy [3] $\pi_t^s(x) := \mathbb{P}(D_t = 1 \mid X_t = x, S = s)$ to maximize an instantaneous utility $R_t(D_t, Y_t)$, possibly subject to certain constraints. An individual is informed of the decision, and subsequently takes actions that may change the qualification $Y_{t+1}$ and features $X_{t+1}$. The latter is used to drive the institute's decision at the next time step. This process is shown in Fig. 1. Note that this model can be viewed as capturing either a randomly selected individual repeatedly going through the decision cycles, or population-wide average when all individuals are subject to the decision cycles. Thus, $\alpha_t^s := \mathbb{P}(Y_t = 1 \mid S = s)$ is the probability of an individual from $\mathcal{G}_s$ qualified at time $t$ at the individual level, while being the *qualification rate* at the group level. One of our primary goals is to study how $\alpha_t^s$ evolves under different (fair) policies.

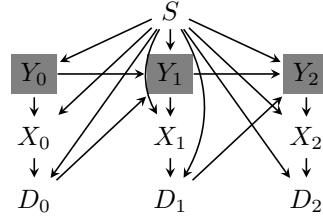

Figure 1: The graphical representation of our model where gray shades indicate latent variables.

**Feature generation process.** In many real-world scenarios, equally qualified individuals from different groups can have different features, potentially due to the different culture backgrounds and physiological differences of different demographic groups. Therefore, we consider that at time step $t$, given $Y_t = y$ and $S = s$, features $X_t$ are generated by $G_y^s(x) := \mathbb{P}(X_t = x \mid Y_t = y, S = s)$. This will be referred to as the *feature distribution* and assumed time-invariant. The convex combination $\mathbb{P}(X_t = x \mid S = s) = \alpha_t^s G_1^s(x) + (1 - \alpha_t^s)G_0^s(x)$ will be referred to as the *composite feature distribution* of group $\mathcal{G}_s$ at time $t$.

**Transition of qualification states.** At time $t$, after receiving decision $D_t$, an individual takes actions such as exerting effort/investment, imitating others, etc., which results in a new qualification $Y_{t+1}$. This is modeled by a set of transitions $T_{yd}^s := \mathbb{P}(Y_{t+1} = 1 \mid Y_t = y, D_t = d, S = s)$, which are time-invariant and group-dependent. These transitions characterize individuals' ability to maintain or improve its qualification. Note that we don't model individuals' strategic responses as in [23, 30], but rather use $T_{yd}^s$ to capture the overall effect; in other words, this single quantity may encapsulate the individual's willingness to exert effort, the cost of such effort, as well as the strength of community support, etc. Specifically, $T_{0d}^s$ (resp. $T_{1d}^s$) represents the probability of individuals from $\mathcal{G}_s$ who were previously unqualified (resp. qualified) became (resp. remain) qualified after receiving decision $d \in \{0, 1\}$. Note that the case when feature distributions or transitions are group-independent is a special case of our formulation, i.e., by setting $G_y^a = G_y^b$ or $T_{yd}^a = T_{yd}^b$.

**Fair myopic policy of an institute.** A myopic policy $\pi_t$ at time $t$ aims at maximizing the instantaneous expected utility/reward $\mathcal{U}(D_t, Y_t) = \mathbb{E}[R_t(D_t, Y_t)]$, where the institute gains $u_+ > 0$ by accepting a qualified individual and incurs a cost $u_- > 0$ by accepting an unqualified individual,

i.e., $R_t(D_t, Y_t) := \begin{cases} u_+, & \text{if } Y_t = 1 \text{ and } D_t = 1 \\ -u_-, & \text{if } Y_t = 0 \text{ and } D_t = 1 \\ 0, & \text{if } D_t = 0 \end{cases}$. A fair myopic policy maximizes the above

utility subject to a fairness constraint $\mathcal{C}$. We focus on a set of group fairness constraints that equalize certain statistical measure between $\mathcal{G}_a$ and $\mathcal{G}_b$. A commonly studied (one-shot) fair machine learning

problem is to find $(\pi_t^a, \pi_t^b)$ that solves the following constrained optimization,

$$\max_{\pi^a, \pi^b} \quad \mathcal{U}(D_t, Y_t) = p_a \mathbb{E}[R_t(D_t, Y_t)|S = a] + p_b \mathbb{E}[R_t(D_t, Y_t)|S = b]$$

$$\text{s.t.} \quad \mathbb{E}_{X_t \sim \mathcal{P}_{\mathcal{C}}^a}[\pi^a(X_t)] = \mathbb{E}_{X_t \sim \mathcal{P}_{\mathcal{C}}^b}[\pi^b(X_t)] , \tag{1}$$

where $\mathcal{P}_{\mathcal{C}}^s$ is some probability distribution over features $X_t$ and specifies the fairness metric $\mathcal{C}$. Many popular fairness metrics can be written in this form, e.g.,

1. Equality of Opportunity (EqOpt) [18]: this requires the true positive rate (TPR) to be equal, i.e., $\mathbb{P}(D_t = 1|Y_t = 1, S = a) = \mathbb{P}(D_t = 1|Y_t = 1, S = b)$. This is equivalent to $\mathbb{E}_{X_t|Y_t=1, S=a}[\pi_t^a(X_t)] = \mathbb{E}_{X_t|Y_t=1, S=b}[\pi_t^b(X_t)]$, i.e., $\mathcal{P}_{\text{EqOpt}}^s(x) = G_1^s(x)$.
2. Demographic Parity (DP) [5]: this requires the positive rate (PR) to be equal, i.e., $\mathbb{P}(D_t = 1|S = a) = \mathbb{P}(D_t = 1|S = b)$. This is equivalent to $\mathbb{E}_{X_t|S=a}[\pi_t^a(X_t)] = \mathbb{E}_{X_t|S=b}[\pi_t^b(X_t)]$, i.e., $\mathcal{P}_{\text{DP}}^s(x) = (1 - \alpha_t^s)G_0^s(x) + \alpha_t^s G_1^s(x)$.

We focus on this class of myopic polices in this paper, and refer to the solution to (1) as the optimal policy. We further define *qualification profile*[4], $\gamma_t^s(x)$, the probability an individual with features $x$ from group $\mathcal{G}_s$ is qualified at $t$, i.e.,

$$\gamma_t^s(x) = \mathbb{P}(Y_t = 1 \mid X_t = x, S = s) = \frac{1}{\frac{G_0^s(x)}{G_1^s(x)}(\frac{1}{\alpha_t^s} - 1) + 1}, \quad x \in \mathbb{R}. \tag{2}$$

Then the utility obtained from the group $\mathcal{G}_s$ at time step $t$ is given by $\mathbb{E}[R_t(D_t, Y_t)|S = s] = \mathbb{E}_{X_t|S=s}[\pi_t^s(X_t)(\gamma_t^s(X_t)(u_+ + u_-) - u_-)]$. Detailed derivation is shown in Appendix E.

## 4 Evolution and Equilibrium Analysis of Qualification Rates

In this section, we first solve the one-shot optimization problem (1) (Sec. 4.1). We then show that under the optimal policy, there exists an equilibrium of qualification rates in the long run, and that a sufficient condition for its uniqueness is also introduced (Sec. 4.2).

### 4.1 Threshold policies are optimal

If an individual's qualification is observable, the optimal policy is straightforward absent of fairness constraints: accepting all qualified ones and rejecting the rest. When qualification is not observable, the institute needs to infer from observed features and accepts those most likely to be qualified. Next we show that under mild assumptions, optimal policies are in the form of threshold policies.

**Assumption 1.** $G_y^s(x)$ and the CDF, $\int_{-\infty}^x G_y^s(z)dz$, are continuous in $x \in \mathbb{R}$, $\forall y, s$; $G_1^s(x)$ and $G_0^s(x)$ satisfy strict monotone likelihood ratio property, i.e., $\frac{G_1^s(x)}{G_0^s(x)}$ is strictly increasing in $x \in \mathbb{R}$.

**Assumption 2.** $\forall s \in \{a, b\}$, $\mathcal{P}_{\mathcal{C}}^s(x)$ is continuous in $x \in \mathbb{R}$; $\frac{\mathbb{P}(X=x|S=s)}{\mathcal{P}_{\mathcal{C}}^s(x)}$ is non-decreasing in $x \in \mathbb{R}$.

Assumption 1 says that an individual is more likely to be qualified as his/her feature value increases[5]. We show that *under Assumption 1, the optimal unconstrained policy is a threshold policy, i.e., $\forall x, t$ and $s \in \{a, b\}$, $\pi_t^s(x) = \mathbf{1}(x \geq \theta_t^s)$ for some $\theta_t^s \in \mathbb{R}$.* Assumption 2 limits the types of fairness constraints, but is satisfied by many commonly used ones, including EqOpt and DP. We show that *for any fairness constraint $\mathcal{C}$ satisfying Assumption 2, the optimal fair policy is a threshold policy.* The proof of these results is given in Appendix F, which is consistent with Theorem 3.2 in [9]. Moreover, under Assumption 1 and 2, a threshold as a function of qualification rates, $\theta_t^s := \theta^s(\alpha_t^a, \alpha_t^b)$, is continuous and non-increasing in $\alpha_t^a$ and $\alpha_t^b$. In the next Lemma 1, we further characterize these optimal (fair) thresholds in the optimal (fair) policies.

**Lemma 1** (Optimal (fair) threshold). *Let $(\gamma^a(x), \gamma^b(x))$ be a pair of qualification profiles for groups $\mathcal{G}_a$ and $\mathcal{G}_b$ at $t$. Let threshold pairs $(\theta_{UN}^{a*}, \theta_{UN}^{b*})$ and $(\theta_{\mathcal{C}}^{a*}, \theta_{\mathcal{C}}^{b*})$ be the unconstrained and fair optimal thresholds under constraint $\mathcal{C}$, respectively. Then we have $\gamma^a(\theta_{UN}^{a*}) = \gamma^b(\theta_{UN}^{b*}) = \frac{u_-}{u_+ + u_-}$ and*

$$p_a\left(\gamma^a(\theta_{\mathcal{C}}^{a*}) - \frac{u_-}{u_+ + u_-}\right)\frac{\mathbb{P}(X = \theta_{\mathcal{C}}^{a*}|S=a)}{\mathcal{P}_{\mathcal{C}}^a(\theta_{\mathcal{C}}^{a*})} + p_b\left(\gamma^b(\theta_{\mathcal{C}}^{b*}) - \frac{u_-}{u_+ + u_-}\right)\frac{\mathbb{P}(X = \theta_{\mathcal{C}}^{b*}|S=b)}{\mathcal{P}_{\mathcal{C}}^b(\theta_{\mathcal{C}}^{b*})} = 0. \tag{3}$$

Here we have removed the subscript $t$ since the thresholds are not $t$-dependent; they only depend on current qualification rates. The solution to Eqn. (3) is the threshold pair $(\theta_\mathcal{C}^{a*}, \theta_\mathcal{C}^{b*})$ that satisfies the fairness constraint $\int_{\theta_\mathcal{C}^{a*}}^\infty \mathcal{P}_\mathcal{C}^a(x)dx = \int_{\theta_\mathcal{C}^{b*}}^\infty \mathcal{P}_\mathcal{C}^b(x)dx$ in Eqn. (1) while maximizing the expected utility $\mathcal{U}(D, Y)$ at time $t$. Under DP and EqOpt fairness, Eqn. (3) can be reduced to

$$p_a\gamma^a(\theta_{\mathtt{DP}}^{a*}) + p_b\gamma^b(\theta_{\mathtt{DP}}^{b*}) = \frac{u_-}{u_+ + u_-}; \quad \frac{p_a\alpha^a}{\gamma^a(\theta_{\mathtt{EqOpt}}^{a*})} + \frac{p_b\alpha^b}{\gamma^b(\theta_{\mathtt{EqOpt}}^{b*})} = \frac{p_a\alpha^a}{\frac{u_-}{u_+ + u_-}} + \frac{p_b\alpha^b}{\frac{u_-}{u_+ + u_-}}.$$

Lemma 1 also indicates the relation between the unconstrained and fair optimal polices, e.g., a group's qualification profile evaluated at the unconstrained threshold is the same as the weighted combination of two groups' qualification profiles evaluated at their corresponding fair thresholds under DP.

## 4.2 Evolution and equilibrium analysis

We next examine what happens as the institute repeatedly makes decisions based on the optimal (fair) policies derived in Sec. 4.1, while individuals react by taking actions to affect their future qualifications. We say the qualification rate of $\mathcal{G}_s$ is at an *equilibrium* if $\alpha_{t+1}^s = \alpha_t^s, \forall t \geq t_o$ for some $t_o$, or equivalently, if $\lim_{t\to\infty} \alpha_t^s = \alpha^s$ is well-defined for some $\alpha^s \in [0, 1]$. Analyzing equilibrium helps us understand the property of the population in the long-run. We begin by characterizing the dynamics of qualification rates $\alpha_t^s$ under policy $\pi_t^s$ as follows (see Appendix E for the derivation):

$$\alpha_{t+1}^s = g^{0s}(\alpha_t^a, \alpha_t^b)\cdot(1 - \alpha_t^s) + g^{1s}(\alpha_t^a, \alpha_t^b)\cdot\alpha_t^s, \quad s \in \{a, b\}, \tag{4}$$

where $g^{ys}(\alpha_t^a, \alpha_t^b) := \mathbb{E}_{X_t|Y_t=y,S=s}\left[(1 - \pi_t^s(X_t))T_{y0}^s + \pi_t^s(X_t)T_{y1}^s\right]$ depends on qualification rates $\alpha_t^a, \alpha_t^b$ through the policy $\pi_t^s$. When $\pi_t^s(x) = \mathbf{1}(x \geq \theta_t^s)$, this reduces to $g^{ys}(\alpha_t^a, \alpha_t^b) := T_{y0}^s \int_{-\infty}^{\theta_t^s} G_y^s(x)dx + T_{y1}^s \int_{\theta_t^s}^\infty G_y^s(x)dx, y \in \{0, 1\}$. Denote $g^{ys}(\alpha_t^a, \alpha_t^b) := g^{ys}(\theta^s(\alpha_t^a, \alpha_t^b))$.

Dynamics (4) says that the qualified people at each time consists of two parts: the qualified people in the previous time step remain being qualified, and those who were unqualified in the previous time step change to become qualified.

Theorem 1 below shows that for any transition and any threshold policy that are continuous in qualification rates, the above dynamical system always has at least one equilibrium.

**Theorem 1** (Existence of equilibrium). *Consider a dynamics* (4) *with a threshold policy $\theta^s(\alpha^a, \alpha^b)$ that is continuous in $\alpha^a$ and $\alpha^b$. $\forall T_{yd}^s \in (0, 1)$, there exists at least one equilibrium $(\widehat\alpha^a, \widehat\alpha^b)$.*

While an equilibrium exists under any set of transitions, its specific property (e.g., quantity, value, etc.) highly depends on transition probabilities which specify different user dynamics. We focus on two scenarios given in the condition below.

**Condition 1.** $\forall s \in \{a, b\}$, *(A)* $T_{01}^s \leq T_{00}^s$ *and* $T_{11}^s \leq T_{10}^s$ ; *(B)* $T_{01}^s \geq T_{00}^s$ *and* $T_{11}^s \geq T_{10}^s$.

As mentioned, transitions $T_{yd}^s$ characterize the ability of individuals from $\mathcal{G}_s$ to maintain/improve their future qualifications, this value summarizes individual's behaviors. On one hand, an accepted individual may feel less motivated to remain qualified (if it was) or become qualified (if it was not). On the other hand, the accepted individual may have access to better resources or feel more inspired to remain or become qualified. These competing factors (referred to later as the "lack of motivation" effect and the "leg-up" effect, respectively) may work simultaneously, and the net effect can be context dependent. Condition 1(A) (resp. Condition 1(B)) suggests that the first (resp. second) effect is dominant for both qualified and unqualified individuals. There are two other combinations: (C) $T_{01}^s \geq T_{00}^s$ and $T_{11}^s \leq T_{10}^s$; (D) $T_{01}^s \leq T_{00}^s$ and $T_{11}^s \geq T_{10}^s$, under which the qualified and unqualified are dominant by different effects. These cases incur more uncertainty; slight changes in feature distributions or transitions may result in opposite conclusions. More discussions are in Appendix D.

Given the existence of an equilibrium, Theorem 2 further introduces sufficient conditions for it to be unique. Based on the unique equilibrium, we can evaluate and compare policies (Sec. 5), and design effective interventions to promote long-term equality and/or the overall qualifications (Sec. 6).

**Theorem 2** (Uniqueness of equilibrium). *Consider a decision-making system with dynamics* (4) *and either unconstrained or fair optimal threshold policy. Let $h^s(\theta^s(\alpha^a, \alpha^b)) := \frac{1 - g^{1s}(\theta^s(\alpha^a, \alpha^b))}{g^{0s}(\theta^s(\alpha^a, \alpha^b))}, s \in \{a, b\}$. Under Assumptions 1 and 2, a sufficient condition for* (4) *to have a unique equilibrium is as follows, $\forall s \in \{a, b\}$:*

1. *Under Condition 1(A),* $\left| \frac{\partial h^s(\theta^s(\alpha^a, \alpha^b))}{\partial \alpha^{-s}} \right| < 1, \forall \alpha^s \in [0, 1],$ *where* $-s := \{a, b\} \setminus s;$
2. *Under Condition 1(B),* $\left| \frac{\partial h^s(\theta^s(\alpha^a, \alpha^b))}{\partial \alpha^{-s}} \right| < 1$ *and* $\left| \frac{\partial h^s(\theta^s(\alpha^a, \alpha^b))}{\partial \alpha^s} \right| < 1, \forall \alpha^a, \alpha^b \in [0, 1].$

These sufficient conditions can further be satisfied if for the qualified ($y = 1$) and the unqualified ($y = 0$), transitions $T_{y1}^s$ and $T_{y0}^s$ are sufficiently close (see Corollary 1, Appendix F), i.e., policies have limited influence on the qualification dynamics. It is worth noting that the conditions of Theorem 2 only guarantee uniqueness of equilibrium but not stability, i.e., it is possible that the qualification rates oscillate and don't converge under this discrete-time dynamics (see examples on COMPAS data in Sec. 7). The uniqueness can be guaranteed and further attained if the dynamics (4) satisfies $L$-Lipschitz condition with $L < 1$. However, Lipschitz condition is relatively stronger than the condition in Theorem 2 (see the comparison in Appendix D).

Figure 2 illustrates trajectories of qualification rates $(\alpha_t^a, \alpha_t^b)$ and the equilibrium for a Gaussian case under Condition 1(B) (see details in Appendix B). Let $g^{ys} := g^{ys}(\theta^s(\alpha^a, \alpha^b))$, the points $(\alpha^a, \alpha^b)$ on the red, and blue dashed curves satisfy $\alpha^b = g^{0b}(1 - \alpha^b) + g^{1b}\alpha^b$ and $\alpha^a = g^{0a}(1 - \alpha^a) + g^{1a}\alpha^a$, respectively.

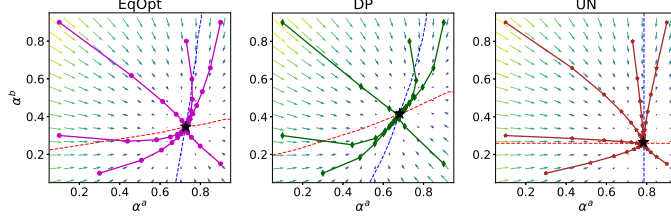

Figure 2: Illustration of $\{(\alpha_t^a, \alpha_t^b)\}_t$ for a Gaussian case under EqOpt, DP, UN optimal policies: each plot shows 6 sample paths with each circle/diamond/star representing one pair of $(\alpha_t^a, \alpha_t^b)$.

Their intersection (black star) is the equilibrium $(\widehat{\alpha}^a, \widehat{\alpha}^b)$. The sufficient conditions in Theorem 2 guarantee these two curves have only one intersection. Moreover, observe that these two curves split the plane $\{(\alpha^a, \alpha^b) : \alpha^a \in [0, 1], \alpha^b \in [0, 1]\}$ into four parts, which can be used for determining how $(\alpha_t^a, \alpha_t^b)$ will change at $t$. For example, if $(\alpha_t^a, \alpha_t^b)$ falls into the left side of the blue dashed curve, then $\alpha_{t+1}^a > \alpha_t^a$; if $(\alpha_t^a, \alpha_t^b)$ falls into the lower side of the red dashed curve, then $\alpha_{t+1}^b > \alpha_t^b$.

## 5 The Long-term Impact of Fairness Constraints

In this section, we analyze the long-term impact of imposing fairness constraints on the equality of group qualification. We will do so in the presence of *natural equality (and inequality)* [37] where equitable equilibria are attained naturally without imposing additional constraints (in our context, this means attaining $\widehat{\alpha}_{\text{UN}}^a = \widehat{\alpha}_{\text{UN}}^b$ using unconstrained polices).

Although there may exist multiple equilibria, in this section we will assume the conditions in Theorem 2 hold under Assumption 1 and 2 and limit ourselves to the unique equilibrium cases under DP and EqOpt, thereby providing a theoretical foundation and an illustration of how their long-term impact can be compared. As shown below, these short-term fairness interventions may not necessarily promote long-term equity, and their impact can be sensitive to feature distributions and transitions. A small change in either can lead to contrarian results, suggesting the importance of understanding the underlying population.

**Long-term impact on natural equality.** When there is natural equality, an unconstrained optimal policy will result in two groups converging to the same qualification rate, thus rendering fairness constraints is unnecessary. The interesting question here is whether applying a fairness constraint can disrupt the equality. The theorem below shows that the DP and EqOpt fairness will do harm if the feature distributions are different.

**Theorem 3.** *For any feature distribution $G_y^s(x)$ and $\forall \alpha_{\text{UN}} \in (0, 1)$, there exist transitions $\{T_{yd}^s\}_{y,d,s}$ satisfying either Condition 1(A) or Condition 1(B) such that $\widehat{\alpha}_{\text{UN}}^a = \widehat{\alpha}_{\text{UN}}^b = \alpha_{\text{UN}}$. In this case, if $G_y^a(x) \neq G_y^b(x)$ (resp. $G_y^a(x) = G_y^b(x)$), then imposing either $\mathcal{C} = \text{DP}$ or EqOpt fair optimal policies will violate (resp. maintain) equality, i.e., $\widehat{\alpha}_{\mathcal{C}}^a \neq \widehat{\alpha}_{\mathcal{C}}^b$ (resp. $\widehat{\alpha}_{\mathcal{C}}^a = \widehat{\alpha}_{\mathcal{C}}^b$).*

Theorem 3 shows that $\forall \alpha_{\text{UN}} \in (0, 1)$, there exists model parameters under which $\alpha_{\text{UN}}$ is the equilibrium and natural equality is attained. Also, natural equality is not disrupted by imposing either fairness constraint when feature distributions are the same across different groups (referred to as *demographic-invariant* below). However, imposing either constraint will lead to unequal outcomes if feature

distributions are diverse across groups (referred to as *demographic-variant* below), which is more likely to happen in reality. Thus, in these natural equality cases, imposing fairness will often do harm.

**Long-term impact on natural inequality.** Natural inequality, i.e., $\widehat{\alpha}_{UN}^a \neq \widehat{\alpha}_{UN}^b$, is more common than natural equality which only occurs under specific model parameters. This difference in qualification rates at equilibrium typically stems from the fact that either feature distributions or transitions or both are different across different groups. Thus, below we study the impact of imposing fairness by considering these two sources of inequality separately, and we aim to examine whether fairness constraints can address the inequality caused by each. Let *disadvantaged group* be the group with a lower qualification rate at equilibrium.

*Demographic-invariant feature distribution with demographic-variant transition.* In this case, we have the same feature distributions but different transitions in different groups, i.e., $G_y^s = G_y^b, T_{yd}^a \neq T_{yd}^b$. A real-world example is college admission based on ACT/SAT scores: given the same qualification state, score distributions may be the same regardless of the applicant's socio-economic status, but the economically advantaged may be able to afford more investments and effort to improve their score after a rejection.

**Theorem 4.** *Under Condition 1(A),* `DP` *and* `EqOpt` *fairness exacerbate inequality, i.e.,* $|\widehat{\alpha}_C^a - \widehat{\alpha}_C^b| \geq |\widehat{\alpha}_{UN}^a - \widehat{\alpha}_{UN}^b|$; *under Condition 1(B),* `DP` *and* `EqOpt` *fairness mitigate inequality, i.e.,* $|\widehat{\alpha}_C^a - \widehat{\alpha}_C^b| \leq |\widehat{\alpha}_{UN}^a - \widehat{\alpha}_{UN}^b|$. *Moreover, the disadvantaged group remains disadvantaged in both cases, i.e.,* $(\widehat{\alpha}_{UN}^a - \widehat{\alpha}_{UN}^b)(\widehat{\alpha}_C^a - \widehat{\alpha}_C^b) \geq 0$.

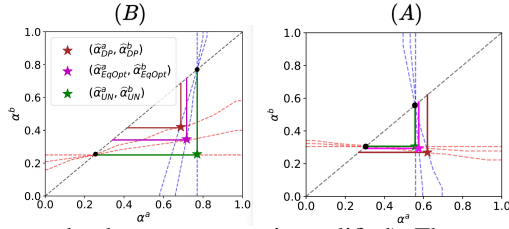

This theorem shows that imposing fairness only helps when the "leg-up" effect is more prominent than the "lack of motivation" effect; alternatively, this suggests that when the "lack of motivation" effect is dominant, imposing fairness should be accompanied by other support structure to dampen this effect (e.g., by helping those accepted to become or remain qualified). Theorem 4 is illustrated in the plot to the right, where transitions satisfy Condition 1(A)-(B) and $G_y^a(x) = G_y^b(x)$ is Gaussian distributed. Each plot includes 3 pairs of red/blue dashed curves corresponding to 3 policies (`EqOpt`, `DP`, `UN`). Points $(\alpha^a, \alpha^b)$ on these curves satisfy $\alpha^b = g^{0b}(\alpha^a, \alpha^b) \cdot (1 - \alpha^b) + g^{1b}(\alpha^a, \alpha^b) \cdot \alpha^b$ and $\alpha^a = g^{0a}(\alpha^a, \alpha^b) \cdot (1 - \alpha^a) + g^{1a}(\alpha^a, \alpha^b) \cdot \alpha^a$, respectively. Each intersection (colored star) is an equilibrium $(\widehat{\alpha}_C^a, \widehat{\alpha}_C^b)$; the length of colored segments represents $|\widehat{\alpha}_C^a - \widehat{\alpha}_C^b|$. The black circle is the intersection of all three blue/red curves.

*Demographic-variant feature distribution with demographic-invariant transition.* In this case, we have the same transitions and different feature distributions in different groups, i.e., $G_y^s \neq G_y^b, T_{yd}^a = T_{yd}^b$. In the same example of college admission this is a case where the ACT/SAT scores are biased against a certain group but there is no difference in how different groups react to the decision. Here, we will focus on a class of feature distributions where those qualified have the same feature distribution regardless of group membership, while those unqualified from $\mathcal{G}_b$ are more likely to have lower features than those unqualified from $\mathcal{G}_a$. This is given in the condition below.

**Condition 2.** $G_y^s(x)$ *is continuous in* $x \in \mathbb{R}$; $G_1^a(x) = G_1^b(x), \forall x \in \mathbb{R}$; $G_0^a(x)$ *and* $G_0^b(x)$ *satisfy strict monotone likelihood ratio property, i.e.,* $\frac{G_0^a(x)}{G_0^b(x)}$ *is strict increasing in* $x \in \mathbb{R}$.

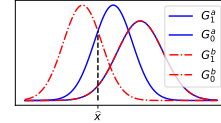

Condition 2 also implies that $\int_{-\infty}^x G_0^b(z)dz \geq \int_{-\infty}^x G_0^a(z)dz, \forall x \in \mathbb{R}$. Let $\widehat{x}$ be defined such that $G_0^b(\widehat{x}) = G_0^a(\widehat{x})$ holds, which is unique. An example satisfying Condition 2 is shown on the right.

**Theorem 5.** *Under Condition 1(B) and Condition 2, if* $\frac{u_+}{u_-} \geq \frac{G_0^s(\widehat{x})}{G_1^s(\widehat{x})} \frac{1 - T_{10}}{T_{00}}$, *we have*

- $\widehat{\alpha}_{UN}^a > \widehat{\alpha}_{UN}^b$ *and* $\widehat{\alpha}_{UN}^a - \widehat{\alpha}_{UN}^b > \widehat{\alpha}_{EqOpt}^a - \widehat{\alpha}_{EqOpt}^b \geq 0$ *hold, i.e.,* `EqOpt` *fairness always mitigates inequality and the disadvantaged group* $\mathcal{G}_b$ *remains disadvantaged.*
- `DP` *fairness may either (1) mitigate inequality, i.e.,* $\widehat{\alpha}_{UN}^a - \widehat{\alpha}_{UN}^b > \widehat{\alpha}_{DP}^a - \widehat{\alpha}_{DP}^b \geq 0$; *or (2) flip the disadvantaged group from* $\mathcal{G}_b$ *to* $\mathcal{G}_a$, *i.e.,* $\widehat{\alpha}_{DP}^b \geq \widehat{\alpha}_{DP}^a$.

Because $\mathcal{G}_a$ and $\mathcal{G}_b$ only differ in $G_0^s(x)$, the condition in Thm 5 ensures at least one group has enough unqualified people to be accepted and can be satisfied if benefit $u_+$ is sufficiently larger than cost $u_-$. We see that in this case the comparison is much more complex depending on the model parameters.

# 6 Effective interventions

As discussed, imposing static fairness constraints is not always a valid intervention in terms of its long-term impact. In some cases it reinforces existing disparity; even when it could work, doing it right can be very hard due to its sensitivity to problem parameters. In this section, we present several alternative interventions that can be more effective in inducing more equitable outcomes or improving overall qualification rates in the long run. We shall assume that the sufficient conditions of Theorem 2 hold under Assumption 1 and 2 so that the equilibrium is unique.

**Policy intervention.** In many instances, preserving static fairness at each time $t$ is important, for short-term violations may result in costly lawsuits [1]. Proposition 1 below shows that using *sub-optimal* fair policies instead of the optimal ones can improve overall qualification in the long run.

**Proposition 1.** *Let* $(\theta_{\mathcal{C}}^a, \theta_{\mathcal{C}}^b)$, $(\theta_{\mathcal{C}}^{a'}, \theta_{\mathcal{C}}^{b'})$ *be thresholds satisfying fairness constraint* $\mathcal{C}$ *under the optimal and an alternative policy, respectively. Let* $(\widehat{\alpha}_{\mathcal{C}}^a, \widehat{\alpha}_{\mathcal{C}}^b)$, $(\widehat{\alpha}_{\mathcal{C}}^{a'}, \widehat{\alpha}_{\mathcal{C}}^{b'})$ *be the corresponding equilibrium.*

- *If* $\theta_{\mathcal{C}}^{s'}(\alpha^a, \alpha^b) > \theta_{\mathcal{C}}^s(\alpha^a, \alpha^b)$, $\forall \alpha^s \in [0, 1]$ *under Condition 1(A), then* $\widehat{\alpha}_{\mathcal{C}}^{s'} > \widehat{\alpha}_{\mathcal{C}}^s$, $\forall s \in \{a, b\}$;
- *If* $\theta_{\mathcal{C}}^{s'}(\alpha^a, \alpha^b) < \theta_{\mathcal{C}}^s(\alpha^a, \alpha^b)$, $\forall \alpha^s \in [0, 1]$ *under Condition 1(B), then* $\widehat{\alpha}_{\mathcal{C}}^{s'} > \widehat{\alpha}_{\mathcal{C}}^s$, $\forall s \in \{a, b\}$.

Note that the sacrifice is in instantaneous utility, not necessarily in total utility in the long run (see an example in proof of Proposition 1, Appendix F). If static fairness need not be maintained at all times, then we can employ separate policies for each group, and Proposition 2 below shows that under certain conditions on transitions, threshold policies leading to equitable equilibrium always exist.

**Proposition 2.** *Let* $\mathcal{I}_s := \left[ \frac{1 - \max\{T_{11}^s, T_{10}^s\}}{\max\{T_{01}^s, T_{00}^s\}}, \frac{1 - \min\{T_{11}^s, T_{10}^s\}}{\min\{T_{01}^s, T_{00}^s\}} \right]$, $s \in \{a, b\}$. *Under Condition 1(A) or 1(B), if* $\mathcal{I}_a \cap \mathcal{I}_b \neq \emptyset$, *then* $\forall \widehat{\alpha} \in \mathcal{I}_a \cap \mathcal{I}_b$, *there exist threshold policies* $\theta^s(\alpha^s)$, $\forall \alpha^s \in [0, 1]$, *under which* $\alpha_t^s \to \widehat{\alpha}$, $\forall s \in \{a, b\}$, *i.e., equitable equilibrium is attained; if* $\mathcal{I}_a \cap \mathcal{I}_b = \emptyset$, *then there is no threshold policy that can result in equitable equilibrium.*

Proposition 2 also indicates that when two groups' transitions are significantly different, manipulating policies cannot achieve equality. In this case, the following intervention can be considered.

**Transition Intervention.** Another intervention is to alter the value of transitions, e.g., by establishing support for both the accepted and rejected. Proposition 3 shows that the qualification rate $\widehat{\alpha}^s$ at equilibrium can be improved by enhancing individuals' ability to maintain/improve qualification, which is consistent with the empirical findings in loan repayment [41, 17, 22] and labor markets [14].

**Proposition 3.** $\forall s \in \{a, b\}$, *increasing any transition probability* $T_{yd}^s$, $d \in \{0, 1\}$, $y \in \{0, 1\}$ *always increases the value of equilibrium qualification rates* $\widehat{\alpha}^s$.

# 7 Experiments

We conducted experiments on both Gaussian synthetic datasets and real-world datasets. We present synthetic data experiments in Appendix B and the results using real-world datasets here. These are static, one-shot datasets, which we use to create a simulated dynamic setting as detailed below.

**Loan repayment study.** We use the FICO score dataset [42] to study the long-term impact of fairness constraints `EqOpt` and `DP` and other interventions on loan repayment rates in the Caucasian group $\mathcal{G}_C$ and the African American group $\mathcal{G}_{AA}$. With the pre-processed data in [18], we simulate a dataset with loan repayment records and credits scores. We first compute the initial qualification (loan repayment)

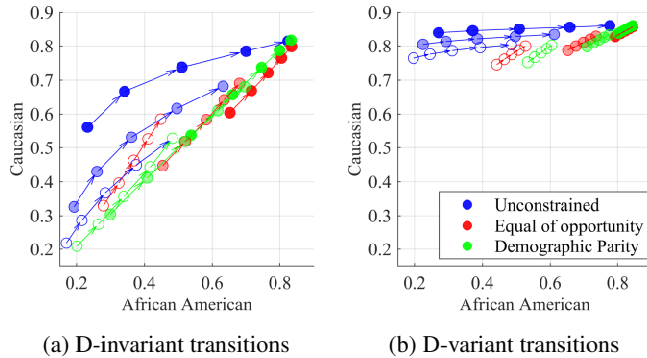

(a) D-invariant transitions    (b) D-variant transitions

Figure 3: Results on the FICO dataset: Points are the equilibria of repayment rates in $\mathcal{G}_{AA}$, $\mathcal{G}_C$ under Condition 1(B) with different transitions. Arrows indicate the direction of increasing $T_{01}^s$; a more transparent point represents the smaller value of $T_{10}^s$. In panel (a), $T_{yd}^{AA} = T_{yd}^C$, while in panel (b), $T_{yd}^{AA} < T_{yd}^C$.

rates and estimate the feature distributions $G_y^s(x)$ with beta distributions based on the simulated data. Then, we compute the optimal UN, EqOpt, DP threshold according to Eqn. (3). Consequently, with the dynamics (4), we update the qualification rates in both groups. This process proceeds and qualification rates in both groups change over time.

Our results show consistent findings with studies in loan repayment literature [17, 41]. Specifically, [41] studied the loan repayment in group lending and pointed out that in practice effective training and leadership among the groups who were issued loans can increase their willingness to pay and improve the group repayment rate. Similar conclusion is also suggested by [17]. In our model, these interventions can be regarded as stimulating transitions (i.e., $T_{y1}$) to improve the future repayment rates. And the scenarios under such intervention would satisfy Condition 1(B). Fig. 3 shows that under Condition 1(B), increasing the transition $T_{01}^s$ always increases qualification rates, and DP in general can result in a more equitable equilibrium than EqOpt. Fig. 3(a) shows that in Demographic-invariant (D-invariant) transition cases ($T_{yd}^{AA} = T_{yd}^C$): (1) $\mathcal{G}_{AA}$ always remains as disadvantaged group; (2) when $T_{10}$ is small, the inequality under UN optimal policies is small and the intervention on $T_{01}$ only has minor effects on equality; when $T_{10}$ is large (darker blue points), varying $T_{01}$ can affect disparity significantly; (3) imposing DP attains equitable equilibria in general, which is robust to transitions and consistent with the conclusion in [37]; (4) when $T_{10}$ is small, imposing EqOpt exacerbates inequality as $T_{01}$ increases; while $T_{10}$ is sufficient large, equality can be attained and robust to transitions. In Fig. 3(b), it shows that in D-variant transition cases, by setting $T_{yd}^{AA} < T_{yd}^C$, the inequality between $\mathcal{G}_{AA}$ and $\mathcal{G}_C$ further gets reinforced. In summary, the effectiveness of such intervention (increasing $T_{01}$) on promoting equality highly depends on the value of $T_{10}$ and policies.

**The COMPAS data.** Our second set of experiments is conducted on a multivariate recidivism prediction dataset from Correctional Offender Management Profiling for Alternative Sanctions (COMPAS) [3]. We again use this static and high-dimensional dataset to create a simulated decision-making process as the FICO experiments. Specifically, from the raw data we calculate the initial qualification (recidivism) rate and train optimal classifier using a logistic regression model, based

Table 1: osi*/osi$_H$/osi$_L$ is the percentage that oscillation occurs among 125 set of different transitions under policy UN*/UN$_{\theta_H}$/UN$_{\theta_L}$. Among transitions that lead to stable equilibrium, Col 2/Col 3 shows the percentage that UN$_{\theta_H}$/UN$_{\theta_L}$ results in lower recidivism compared with UN*.

|   | $\widehat{\alpha}_{\theta_H} < \widehat{\alpha}^*$ | $\widehat{\alpha}_{\theta_L} < \widehat{\alpha}^*$ | osi* | osi$_H$ | osi$_L$ |
|---|---|---|---|---|---|
| A | 0 | 1 | 0.29 | 0.12 | 0.36 |
| B | 0.99 | 0.01 | 0 | 0 | 0 |
| C | 0.37 | 0.28 | 0 | 0 | 0 |
| D | 0.79 | 0.63 | 0.06 | 0 | 0.13 |

on which recidivism rate is updated according to Eqn. (4) under a given set of transitions. In the context of recidivism prediction, we consider all the possible types of transitions under an unconstrained policy, i.e., transitions satisfying conditions 1(A)-(D). The classifier decision here corresponds to incarceration based on predicted likelihood of recidivism: the higher the predicted recidivism, the more likely an incarceration decision. In subsequent time steps, the data is re-sampled from the raw data proportional to the updated recidivism rates. This process repeats and the group recidivism rates change over time. Our results here primarily serve to highlight the complexity in such a decision-making system. In particular, we see that an equilibrium may not exist and under some transitions the qualification rate may oscillate. Specifically, Table 1 shows that Prop. 1 holds under Condition 1(A)-(B); there is no oscillation under Condition 1(B)-(C); under Condition 1(C)-(D), there is more uncertainty which is discussed in Appendix D. More results on the oscillation can be found in Appendix B.

## 8 Conclusion

In this paper, we studied the long-term impact of fairness constraints (e.g., DP and EqOpt) on group qualification rates. By casting the problem in a POMDP framework, we conducted equilibrium analysis. Specifically, we first identified sufficient conditions for the existence and uniqueness of equilibrium, under which we compared different fairness constraints regarding their long-term impact. Our findings show that the same fairness constraint can have opposite impact depending on the underlying problem scenarios, which highlights the importance of understanding real-world dynamics in decision making systems. Our experiments on real-world static datasets with simulated dynamics also show that our framework can be used to facilitate social science studies. Our analysis has focused on scenarios with a unique equilibrium; scenarios with multiple equilibria or oscillating states remain an interesting direction of future research.

## Acknowledgement

X. Zhang, Y. Liu and M. Liu have been supported by the NSF under grants CNS-1616575, CNS-1646019, CNS-1739517, IIS-2007951, and by the ARO under contract W911NF1810208. R. Tu would like to acknowledge the funding support of the Swedish e-Science Research Centre and the material suggestion regarding the social impact of polices given by Yating Zhang. Part of the work was done when R. Tu was a visiting student in Microsoft Research, Cambridge, and he would like to acknowledge Microsoft's travel support. K. Zhang would like to acknowledge the support by the United States Air Force under Contract No. FA8650-17-C-7715.

## Broader Impact

In this paper, we focus on the (un)fairness issue that arises in automated decision-making systems and aim to understand the long-term impact of algorithmic (fair) decisions on the well-being of different sub-groups in a population. Our partially observed sequential decision making framework is applicable to a wide range of domains (e.g., lending, recruitment, admission, criminal justice, etc.). By conducting an equilibrium analysis and evaluating the long-term impact of different fairness criteria, our results provide a theoretical foundation that can help answer questions such as whether/when imposing short-term fairness constraints are effective in promoting long-term equality.

First of all, our results can help policymakers (e.g., companies, banks, governments, etc.) in their decision making process by highlighting the potential pitfalls of commonly used static fairness criteria and providing guidance on how to design effective interventions that can avoid such unintended consequences and result in positive long-term societal impacts.

Secondly, our results may be useful to research in fields outside of the computer science community. The experiments on static real-world datasets have shown consistent findings with literature in social sciences [36, 17, 41]. Although these empirical results are obtained using simulated dynamics due to a lack of real datasets, they may provide insights and theoretical supports for research in other fields.

Lastly, while this work is limited to binary decisions, the main take-away can be applied in other applications such as computer vision, natural language processing, etc., using more complicated classifiers such as DNN. We hope that our work will encourage researchers in these domains to similarly consider discrimination risks when developing techniques, and raise awareness that static fairness constraint may not suffice and long-term fairness cannot be designed in a vacuum without considering the human element. We thus emphasize the importance of performing real-time measurements and developing proper fair classifiers from dynamic datasets.

Having mentioned the potential positive impact of our work, we also want to point out the limitations in our model and analysis. Firstly, in this work we use a set of transitions $T_{yd}^s$ to capture individuals' abilities to improve/maintain future qualifications, and our analysis and conclusions rely on this set of values. In practice, however, these quantities can be extremely hard to measure due to the complexity of human behaviors and environmental factors. In addition, as we have noted in the paper, in some cases the conclusion can be highly sensitive to minor changes in these transitions. Secondly, our theoretical results have focused on scenarios with a unique equilibrium, while in practice the situation can be much more complicated (multiple equilibria or no equilibria), as demonstrated by the oscillations we see in the COMPAS simulation study. Thus, it is worthwhile for future work to consider these more complex cases. Lastly, due to the lack of dynamic datasets, our experiments are performed over static real-world datasets with simulated dynamics. Thus, an accurate model of real-world dynamics is needed when deploying our method for practical decision making.

## Footnotes

[2]For simplicity of exposition, our analysis is based on one-dimensional feature space. However, the conclusions hold for high-dimensional features. This can be done by first mapping the feature space to a one-dimensional qualification profile space, this extension is given in Appendix C.

[3]We use group-dependent policies so that the optimal policies can achieve the *perfect* fairness, i.e., certain statistical measures are equalized *exactly*, which allows us to study the impact of fairness constraint *precisely*. Although using group-dependent policies might be prohibited in some scenarios (e.g., criminal justice), our qualitative conclusions are applicable to cases when two groups share the same policy, under which the *approximate* fairness is typically attained to maximize utility.

[4]We assume the institute has perfect knowledge of $\gamma_t^s(x)$. In practice, this is obtained via learning/estimating $\alpha_t^s$ and $G_y^s(x)$ from data [25, 40].

[5]When qualification increases as the feature value $x$ decreases, one can simply use the opposite of $x$.

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
