[Supplementary Material · appendix.pdf]

## A Notations

| | |
|---|---|
| $\mathcal{G}_s$ | demographic group, $s \in \{a, b\}$ |
| $X_t$ | feature at $t$, $X \in \mathbb{R}^d$ |
| $Y_t$ | true qualification state at $t$, $Y_t \in \{0, 1\}$ |
| $S$ | sensitive attribute $S \in \{a, b\}$ |
| $p_s$ | group proportion of $s$, i.e., $p_s = \mathbb{P}(S = s)$ |
| $D_t$ | institute's decision at $t$, $D_t \in \{0, 1\}$ |
| $\pi_t^s(x)$ | policy for $\mathcal{G}_s$ at $t$, i.e., $\pi_t^s(x) = \mathbb{P}(D_t = 1 \mid X_t = x, S = s)$ |
| $G_y^s(x)$ | feature distribution of unqualified ($y = 0$) or qualified ($y = 1$) people from $\mathcal{G}_s$ , i.e., $\mathbb{P}(X_t = x \mid Y_t = y, S = s)$ |
| $\mathbb{G}_y^s(x)$ | CDF of $G_y^s(x)$, i.e., $\mathbb{G}_y^s(x) = \int_{-\infty}^x G_y^s(z)dx$ |
| $\mathcal{P}_{\mathcal{C}}^s(x)$ | a probability distribution over $X_t$ that specifies the fairness metric $\mathcal{C}$ |
| $\alpha_t^s$ | qualification rate of $\mathcal{G}_s$ at $t$, i.e., $\mathbb{P}(Y_t = 1 \mid S = s)$ |
| $\gamma_t^s(x)$ | qualification profile of $\mathcal{G}_s$ at $t$, i.e., $\mathbb{P}(Y_t = 1 \mid X_t = x, S = s)$ |
| $T_{yd}^s$ | transition probability of $\mathcal{G}_s$, i.e., $\mathbb{P}(Y_{t+1} = 1 \mid Y_t = y, D_t = d, S = s)$ |
| $u_+$ | benefit the institute gains by accepting a qualified individual |
| $u_-$ | cost incurred to the institute by accepting an unqualified individual |
| $\theta_{\mathcal{C}}^s$ | threshold in a threshold policy for $\mathcal{G}_s$ under constraint $\mathcal{C}$, i.e., $\pi_t^s(x) = \mathbf{1}(x \geq \theta_{\mathcal{C}}^s)$ |
| $\widehat{\alpha}_{\mathcal{C}}^s$ | qualification rate of $\mathcal{G}_s$ at the equilibrium under policy with constraint $\mathcal{C} \in \{\texttt{UN}, \texttt{DP}, \texttt{EqOpt}\}$ |

## B Additional results on experiments

**Gaussian distributed synthetic data.** We first verify the conclusions in Section 4 and 5 using the synthetic data, where $X_t \mid Y_t = y, S = s \sim \mathcal{N}(\mu_y^s, (\sigma^s)^2)$.

In Section 4, Figure 2 illustrates sample paths of $\{(\alpha_t^a, \alpha_t^b)\}_t$ under $\texttt{EqOpt}$, $\texttt{DP}$, $\texttt{UN}$ optimal policies. The specific parameters are as follows: $[\mu_0^a, \mu_1^a, \mu_0^b, \mu_1^b] = [-5, 5, -5, 5]$, $[\sigma^a, \sigma^b] = [5, 5]$, $\frac{u_+}{u_-} = 1$, $p_a = p_b = 0.5$, $[T_{00}^a, T_{01}^a, T_{10}^a, T_{11}^a] = [0.4, 0.5, 0.5, 0.9]$, $[T_{00}^b, T_{01}^b, T_{10}^b, T_{11}^b] = [0.1, 0.5, 0.5, 0.7]$.

Table 2 and 3 illustrate the impacts of $\texttt{EqOpt}$ and $\texttt{DP}$ fairness on the equilibrium, where each column shows the value of $\widehat{\alpha}_{\mathcal{C}}^a - \widehat{\alpha}_{\mathcal{C}}^b$ when $\mathcal{C} = \texttt{UN}, \texttt{EqOpt}, \texttt{DP}$ under different sets of parameters. Specifically, in Table 2, $p_a = p_b = 0.5$, $\frac{u_+}{u_-} = 1$, $[\mu_0^s, \mu_1^s, \sigma^s] = [-5, 5, 5], \forall s \in \{a, b\}$ and transitions satisfying either Condition 1(A) or 1(B) are randomly generated; in Table 3, transitions satisfying Condition 1(B) and $G_y^s(x)$ that satisfy Condition 2 are randomly generated, $\frac{u_+}{u_-}$ also satisfies the condition in Theorem 5. These results are consistent with Theorem 4 and 5.

Table 2: $\widehat{\alpha}_{\mathcal{C}}^a - \widehat{\alpha}_{\mathcal{C}}^b$ when $\mathcal{C} = \texttt{UN}, \texttt{EqOpt}, \texttt{DP}$: $G_y^a(x) = G_y^b(x)$ and $T_{yd}^a \neq T_{yd}^b$.

| | Condition 1(A) | | | | | | | | |
|---|---|---|---|---|---|---|---|---|---|
| UN ($\times 10^{-2}$) | -18.45 | 16.89 | 19.82 | -7.21 | -16.34 | -26.56 | 16.66 | -6.03 | -38.63 |
| EqOpt ($\times 10^{-2}$) | -21.11 | 19.13 | 21.78 | -7.62 | -18.56 | -29.21 | 18.14 | -6.28 | -41.52 |
| DP ($\times 10^{-2}$) | -27.98 | 23.11 | 25.65 | -8.90 | -23.11 | -33.22 | 21.09 | -6.66 | -43.35 |
| | Condition 1(B) | | | | | | | | |
| UN ($\times 10^{-2}$) | -19.05 | 18.18 | -0.70 | -58.80 | -40.91 | 61.30 | 12.82 | -44.67 | 2.66 |
| EqOpt ($\times 10^{-2}$) | -18.40 | 17.98 | -0.64 | -57.62 | -34.50 | 48.66 | 12.35 | -41.43 | 2.61 |
| DP ($\times 10^{-2}$) | -17.52 | 17.73 | -0.57 | -55.62 | -28.97 | 36.10 | 11.69 | -37.97 | 2.57 |

Table 3: $\widehat{\alpha}_{\mathcal{C}}^a - \widehat{\alpha}_{\mathcal{C}}^b$ when $\mathcal{C} = \texttt{UN}, \texttt{EqOpt}, \texttt{DP}$: $G_y^a(x) \neq G_y^b(x)$ and $T_{yd}^a = T_{yd}^b$ under Condition 1(B).

| UN ($\times 10^{-2}$) | 1.88 | 26.35 | 2.12 | 0.38 | 5.64 | 12.35 | 11.70 | 0.20 | 4.12 |
|---|---|---|---|---|---|---|---|---|---|
| EqOpt ($\times 10^{-2}$) | 0.57 | 17.43 | 1.75 | 0.32 | 5.05 | 7.81 | 7.21 | 0.18 | 1.68 |
| DP ($\times 10^{-4}$) | 16.26 | 18.29 | -5.94 | -0.93 | -2.25 | 1.47 | 0.92 | -1.68 | -0.80 |

| (a) D-invariant transitions | (b) D-variant transitions |

Figure 5: Results on the FICO dataset: Points are the repayment rates of $\mathcal{G}_{AA}, \mathcal{G}_C$ at the equilibria under Condition 1(B) with different sets of transitions. Arrows indicate the direction of increasing $T_{01}^s$; a more transparent point represents the smaller value of $T_{10}^s$. In panel (a), $T_{yd}^{AA} = T_{yd}^C$, while in panel (b), $T_{yd}^{AA} < T_{yd}^C$.

**FICO score data.** From the pre-processed FICO dataset, we got $\mathbb{P}(X = x \mid S = s)$ and $\mathbb{P}(Y = 1 \mid X = x, S = s)$. In this experiment, we consider two demographic groups, 12% the African American $\mathcal{G}_{AA}$ and 88% the Caucasian $\mathcal{G}_C$. According to the empirical feature distributions, we can first simulate the FICO dataset with credit scores $X$, repayment $Y$, and sensitive attribute $S$. We then compute the initial qualification (repayment) rates $(\alpha_0^{AA}, \alpha_0^C)$, which is 0.34 in $\mathcal{G}_{AA}$ and 0.76 in $\mathcal{G}_C$; and fit Beta distributions to get the feature distribution $\mathbb{P}(X = x \mid S = s, Y = y)$, as shown in Fig. 4. Since the feature distributions are the Beta distributions, we can compute optimal `UN`, `EqOpt`, `DP` thresholds directly using Eqn. (3) and update the repayment rates based on dynamics (4). This process proceeds and $(\alpha_t^{AA}, \alpha_t^C)$ changes over time.

Figure 4: The feature distributions: the scores are rescaled so that they are between 0 and 1.

We then consider the demographic-invariant (D-invariant) and demographic-variant (D-variant) transitions and examine the impact of the transition interventions. Specifically, in the context of loan repayment prediction and group lending [41], the transitions would satisfy Condition 1(B). Fig. 5 illustrates the equilibria $(\widehat{\alpha}^{AA}, \widehat{\alpha}^C)$ under different sets of transitions. Their specific values are listed as follows, where the system has an equilibrium in all cases.

D-invariant: $\quad T_{00} = 0.1, T_{11} = 0.9, \quad T_{10}, T_{01} \in \{0.1, 0.3, 0.5, 0.7, 0.9\}$

D-variant: $\quad T_{00}^{AA} = 0.1, T_{11}^{AA} = 0.9, \quad T_{10}^{AA}, T_{01}^{AA} \in \{0.20, 0.36, 0.53, 0.69, 0.85\}$

$\quad\quad\quad\quad\quad T_{00}^C = 0.4, T_{11}^C = 0.9, \quad T_{10}^C, T_{01}^C \in \{0.45, 0.55, 0.65, 0.75, 0.85\}$

**COMPAS data.** The COMPAS dataset is a high-dimensional dataset with mixed data types (e.g., continuous, binary, and categorical). The number of samples is 5278. There are 10 features and two demographic groups: 60% African American $(\mathcal{G}_{AA})$ and 40% Caucasian $(\mathcal{G}_C)$. The qualification rate in COMPAS is the recidivism rate. The initial recidivism rates are 52.3% in $\mathcal{G}_{AA}$ and 39.1% in $\mathcal{G}_C$.

Due to the complexity of the feature distribution, the system can be either in the equilibrium state or oscillate between two recidivism rates in the long-run. Since the feature distribution is fixed and approximated from the COMPAS dataset, we investigate that under which transitions, the system is in an equilibrium state under unconstrained optimal policy. For this purpose, it is sufficient to study the demographic-invariant transitions $T^{AA} = T^C$ and consider the entire population without distinguishing two groups; moreover, in the context of recidivism prediction, the transitions would satisfy Condition 1 (A). Therefore, we consider $T_{00}$ and $T_{10}$ taking the values 0.1, 0.3, 0.5, 0.7 and 0.9. Figure 6 shows the results when $T_{01} = k \times T_{00}$ and $T_{11} = k \times T_{10}$. We find that when Corollary 1 is satisfied, e.g., when $k \geq 0.5$, most of the corresponding systems have a unique equilibrium (blue

Figure 6: The oscillation level of recidivism rates in the long run is represented by the size of red circles, of which the bigger one represents severer oscillation. The blue dots represent the scenarios with a unique equilibrium. $T_{00}$ and $T_{10}$ axes represent their values respectively; $k$ axis represents the scalar $k$, where $T_{01} = k \times T_{00}$ and $T_{11} = k \times T_{10}$.

(a) $T_{01} = 0.1 \times T_{00}$.

(b) $T_{01} = 0.3 \times T_{00}$.

(c) $T_{01} = 0.5 \times T_{00}$.

(d) $T_{01} = 0.7 \times T_{00}$.

(e) $T_{01} = 0.9 \times T_{00}$.

Figure 7: The oscillation level of recidivism rates under different transitions. In each panel, scalar $k$ denotes the ratio, of which $T_{11} = k \times T_{10}$.

dot). Moreover, when $T_{00} \leq 0.5$, the system is also mostly in the unique equilibrium state. For the other transitions, the system oscillates between two states (red circle). We also show the results under all the combinations of $T_{01}$ and $T_{11}$ in Figure 7.

Next, we study the impact of policy interventions in cases with equilibrium. We randomly choose the transitions under which the system has an equilibrium and then apply the unconstrained policy with optimal threshold (classifier threshold $0.5$), a higher and a lower threshold (classifier thresholds $0.8$ and $0.2$ respectively) compared to the optimum respectively. The results are show in Table 4.

Table 4: Recidivism rates in the long run. UN*: unconstrained policy (UN) with the optimal threshold; $\text{UN}_{\theta_H}$: UN with a higher threshold; $\text{UN}_{\theta_L}$: UN with a lower threshold.

|  | UN* | $\text{UN}_{\theta_H}$ | $\text{UN}_{\theta_L}$ |
|---|---|---|---|
| $\widehat{\alpha}_1$ | 0.164 | 0.166 | 0.147 |
| $\widehat{\alpha}_2$ | 0.343 | 0.356 | 0.307 |
| $\widehat{\alpha}_3$ | 0.230 | 0.246 | 0.162 |
| $\widehat{\alpha}_4$ | 0.306 | 0.3415 | 0.156 |
| $\widehat{\alpha}_5$ | 0.162 | 0.166 | 0.140 |

## C  Generalization to high-dimensional feature space

All analysis and conclusions in this paper can be generalized to high-dimensional feature space $x \in \mathbb{R}^d$, where the qualification profile of $\mathcal{G}_s$ is defined as $\gamma_t^s(\mathbf{x}) = \mathbb{P}(Y_t = 1 \mid \mathbf{X}_t = \mathbf{x}, S = s) \in [0, 1]$, $\mathbf{x} \in \mathbb{R}^d$. Different from one-dimensional case where decisions are made based on features, here decisions are made based on $\gamma_t^s(\mathbf{x})$, i.e., high-dimensional features are mapped into a one-dimensional space first and decisions are made in this transformed space. The threshold policy in this case becomes $\pi_t^s(\mathbf{x}) = \mathbf{1}(\gamma_t^s(\mathbf{x}) \geq \theta_t^s)$ with threshold $\theta_t^s \in [0, 1]$. Let $\gamma_t^{s^{-1}}(\theta) \subset \mathbb{R}^d$ be defined as the preimage of $\theta$ under qualification profile $\gamma_t^s$, then all analysis in one-dimensional settings can be adjusted using $\gamma_t^{s^{-1}}(\cdot)$. For example, Assumption 1 in high-dimensional case can be adjusted to the following: $\forall s \in \{a, b\}$, *given any two thresholds* $0 \leq \theta_j^s < \theta_k^s \leq 1$, *we have* $\gamma_t^{s^{-1}}(K^s) \subset \gamma_t^{s^{-1}}(J^s)$, *where* $J^s = \{\theta : \theta \in [\theta_j^s, 1]\}$ *and* $K^s = \{\theta : \theta \in [\theta_k^s, 1]\}$; in other words, if an individual can get accepted by a policy with the higher threshold, it must be accepted if a policy with a lower threshold was used. Note that this assumption is still mild and always hold if $G_y^s(\mathbf{x})$ belongs to exponential family.

Specifically, if $\forall s \in \{a, b\}, \forall y \in \{0, 1\}$, distribution of $X|Y = y, S = s$ belongs to exponential family and can be written as $G_y^s(\mathbf{x}) := B(\mathbf{x}) \exp \left( \langle \eta(\omega_y^s), \xi(\mathbf{x}) \rangle - A(\omega_y^s) \right)$ for some functions $B(\cdot), \eta(\cdot), \xi(\cdot), A(\cdot)$, where $\langle \mathbf{x}, \mathbf{y} \rangle$ represents inner product of two vectors $\mathbf{x}, \mathbf{y}$ and $\omega_y^s$ is the parameter. Then $\frac{G_0^s(\mathbf{x})}{G_1^s(\mathbf{x})} = \exp \left( - \langle \eta^s, \xi(\mathbf{x}) \rangle + A^s \right)$ where $\eta^s := \eta(\omega_1^s) - \eta(\omega_0^s)$ and $A^s := A(\omega_1^s) - A(\omega_0^s)$. Then

$$\gamma_t^{s^{-1}}(J^s) = \{\mathbf{x} : \gamma_t^s(\mathbf{x}) \geq \theta_j^s\} = \{\mathbf{x} : \langle \eta^s, \xi(\mathbf{x}) \rangle \geq A^s + \log \left( \frac{\frac{1}{\alpha_t^s} - 1}{\frac{1}{\theta_j^s} - 1} \right)\}$$

If $\theta_j^s < \theta_k^s$, then $\log \left( \frac{\frac{1}{\alpha_t^s} - 1}{\frac{1}{\theta_j^s} - 1} \right) < \log \left( \frac{\frac{1}{\alpha_t^s} - 1}{\frac{1}{\theta_k^s} - 1} \right)$. We have $\gamma_t^{s^{-1}}(K^s) \subset \gamma_t^{s^{-1}}(J^s)$.

## D  Discussions

**Transitions under Condition 1(C) or 1(D).**  This paper mainly focus on transitions satisfying Condition 1(A) and 1(B). As mentioned in Section 4.2, there are the other two combinations: (C) $T_{01}^s \geq T_{00}^s$ and $T_{11}^s \leq T_{10}^s$; (D) $T_{01}^s \leq T_{00}^s$ and $T_{11}^s \geq T_{10}^s$, in which there is more uncertainty when conducting equilibrium analysis. The slight changes in the feature distributions or the values of transitions may change conclusions significantly.

Because the system has equilibrium if there is solution to balanced equations defined as Eqn. (5) in Appendix F, i.e., $\frac{1}{\alpha^s} - 1 = \frac{1 - g^{1s}(\theta^s(\alpha^a, \alpha^b))}{g^{0s}(\theta^s(\alpha^a, \alpha^b))}$, $\forall s \in \{a, b\}$. Since

$$\frac{1 - g^{1s}(\theta^s(\alpha^a, \alpha^b))}{g^{0s}(\theta^s(\alpha^a, \alpha^b))} = \frac{1 - \left(T_{10}^s \mathbb{G}_1^s(\theta^s(\alpha^a, \alpha^b)) + T_{11}^s \left(1 - \mathbb{G}_1^s(\theta^s(\alpha^a, \alpha^b))\right)\right)}{T_{00}^s \mathbb{G}_0^s(\theta^s(\alpha^a, \alpha^b)) + T_{01}^s \left(1 - \mathbb{G}_0^s(\theta^s(\alpha^a, \alpha^b))\right)}.$$

Under optimal (fair) policies and Condition 1(A) or 1(B), $\frac{1 - g^{1s}(\theta^s(\alpha^a, \alpha^b))}{g^{0s}(\theta^s(\alpha^a, \alpha^b))}$ is guaranteed to be either decreasing or increasing in $\alpha^s$. This monotonicity is critical to determine the properties (e.g., uniqueness, quantity, value, etc.) of the consequent equilibrium $(\widehat{\alpha}_{\mathcal{C}}^a, \widehat{\alpha}_{\mathcal{C}}^b)$ so that impacts of different fairness can be compared. In contrast, under Condition 1(C) or 1(D), $\frac{1 - g^{1s}(\theta^s(\alpha^a, \alpha^b))}{g^{0s}(\theta^s(\alpha^a, \alpha^b))}$ is no longer monotonic, and its intersection with function $\frac{1}{\alpha^s} - 1$, i.e., equilibrium, is thus hard to characterize. As a consequence, the impacts of different fairness constraints cannot be compared in general.

**Comparison between sufficient conditions in Theorem 2 and Lipschitz condition.**  Let a pair of qualification rats of $\mathcal{G}_a, \mathcal{G}_b$ be noted as $\alpha := (\alpha^a, \alpha^b) \in [0, 1] \times [0, 1]$, and let mapping $\Phi : [0, 1] \times [0, 1] \to [0, 1] \times [0, 1]$ be defined such that dynamical system (4) can be written as $\alpha_{t+1} = \Phi(\alpha_t)$. Then this dynamical system has an equilibrium $\widehat{\alpha}$ if $\Phi(\widehat{\alpha}) = \widehat{\alpha}$. According to Banach Fixed Point Theorem, such equilibrium exists and is unique if the mapping $\Phi$ satisfies $L$-Lipschitz condition with $L < 1$, i.e., $\Phi$ is a contraction mapping. Specifically, $d(\Phi(\alpha_0), \Phi(\alpha_1)) \leq Ld(\alpha_0, \alpha_1)$ for some distance function $d$ and Lipschitz constant $L < 1$.

While Lipschitz condition also ensures the uniqueness of equilibrium, the sufficient conditions given in Theorem 2 are weaker. Use unconstrained optimal policies as an example, in this case dynamics of two groups can be decoupled because threshold $\theta^s(\alpha^a, \alpha^b)$ used in $\mathcal{G}_s$ is independent of qualification of the other group $\alpha^{-s}$. Therefore, sufficient condition $|\frac{\partial h^s(\theta^s(\alpha^a,\alpha^b))}{\partial \alpha^{-s}}| = 0 < 1$ under Condition 1(A) always holds. In contrast, for dynamics of $\mathcal{G}_s$ after decoupling $\alpha_{t+1}^s = \Phi^s(\alpha_t^s) = g^{0s}(\theta^s(\alpha_t^s))(1 - \alpha_t^s) + g^{1s}(\theta^s(\alpha_t^s))\alpha_t^s$, $\Phi^s$ is not necessarily a contraction mapping.

Although sufficient conditions in Theorem 2 are weaker, they do not guarantee the stability of the equilibrium. In contrast, Lipschitz condition with $L < 1$ ensures the unique equilibrium is also stable, i.e., we have $(\alpha_t^a, \alpha_t^b) \to (\widehat{\alpha}^a, \widehat{\alpha}^b)$ given an arbitrary initial state $(\alpha_0^a, \alpha_0^b)$.

## E  Derivations

**Qualification profile of a group.**

$$
\begin{aligned}
\gamma_t^s(x) &= \mathbb{P}(Y_t = 1 | X_t = x, S = s) = \frac{1}{\frac{\mathbb{P}(X_t=x, Y_t=0, S=s)}{\mathbb{P}(X_t=x, Y_t=1, S=s)} + 1} \\
&= \frac{1}{\frac{\mathbb{P}(X_t=x|Y_t=0,S=s)\mathbb{P}(Y_t=0|S=s)}{\mathbb{P}(X_t=x|Y_t=1,S=s)\mathbb{P}(Y_t=1|S=s)} + 1} \\
&= \frac{1}{\frac{\mathbb{P}(X_t=x|Y_t=0,S=s)}{\mathbb{P}(X_t=x|Y_t=1,S=s)}(\frac{1}{\mathbb{P}(Y_t=1|S=s)} - 1) + 1} \\
&= \frac{1}{\frac{G_0^s(x)}{G_1^s(x)}(\frac{1}{\alpha_t^s} - 1) + 1}.
\end{aligned}
$$

**Utility of an institute.**

$$
\mathcal{U}(D_t, Y_t) = \mathbb{E}[R_t(D_t, Y_t)] = \mathbb{P}(S = a)\mathbb{E}[R_t(D_t, Y_t)|S = a] + \mathbb{P}(S = b)\mathbb{E}[R_t(D_t, Y_t)|S = b]
$$

Under policy $\pi^s$, we have

$$
\begin{aligned}
\mathbb{E}[R_t(D_t, Y_t)|S = s] &= \mathbb{P}(D_t = 1, Y_t = 1|S = s)u_+ - \mathbb{P}(D_t = 1, Y_t = 0|S = s)u_- \\
&= \int_x \Big(\mathbb{P}(D_t = 1, Y_t = 1, X_t = x|S = s)u_+ - \mathbb{P}(D_t = 1, Y_t = 0, X_t = x|S = s)u_-\Big)dx \\
&= \int_x \mathbb{P}(X_t = x|S = s)\Big(\mathbb{P}(D_t = 1 | X_t = x, S = s)\mathbb{P}(Y_t = 1 | X_t = x, S = s)u_+ \\
&\quad -\mathbb{P}(D_t = 1 | X_t = x, S = s)\mathbb{P}(Y_t = 0 | X_t = x, S = s)u_-\Big)dx \\
&= \int_x \mathbb{P}(X_t = x|S = s)\Big(\pi^s(x)\gamma_t^s(x)u_+ - \pi^s(x)(1 - \gamma_t^s(x))u_-\Big)dx \\
&= \mathbb{E}_{X_t|S=s}[\pi^s(X_t)(\gamma_t^s(X_t)(u_+ + u_-) - u_-)].
\end{aligned}
$$

Therefore,

$$
\mathcal{U}(D_t, Y_t) = p_a\mathbb{E}_{X_t|S=a}[\pi^a(X_t)(\gamma_t^a(X_t)(u_+ + u_-) - u_-)] + p_b\mathbb{E}_{X_t|S=b}[\pi^b(X_t)(\gamma_t^b(X_t)(u_+ + u_-) - u_-)]
$$

**Dynamics of qualification rate.**

$$
\begin{aligned}
\alpha_{t+1}^s &= \mathbb{P}(Y_{t+1} = 1 \mid S = s) = \int_x \sum_{y,a} \mathbb{P}(Y_{t+1} = 1, Y_t = y, D_t = d, X_t = x \mid S = s)dx \\
&= \int_x \sum_{y,a} \mathbb{P}(Y_{t+1} = 1 \mid Y_t = y, X_t = x, D_t = d, S = s)\mathbb{P}(D_t = d \mid X_t = x, S = s) \\
&\quad \mathbb{P}(X_t = x \mid Y_t = y, S = s)\mathbb{P}(Y_t = y \mid S = s)dx \\
&= \int_x \sum_a \Big\{ \mathbb{P}(Y_{t+1} = 1 \mid Y_t = 0, X_t = x, D_t = d, S = s) \\
&\quad \mathbb{P}(D_t = d \mid X_t = x, S = s)\mathbb{P}(X_t = x \mid Y_t = 0, S = s) \Big\} \mathbb{P}(Y_t = 0 \mid S = s)dx \\
&\quad + \int_x \sum_d \Big\{ \mathbb{P}(Y_{t+1} = 1 \mid Y_t = 1, X_t = x, D_t = d, S = s) \\
&\quad \mathbb{P}(D_t = d \mid X_t = x, S = s)\mathbb{P}(X_t = x \mid Y_t = 1, S = s) \Big\} \mathbb{P}(Y_t = 1 \mid S = s)dx \\
&= \mathbb{E}_{X_t|Y_t=0,S=s}\Big[ (1 - \pi_t^s(X_t))T_{00}^s + \pi_t^s(X_t)T_{01}^s \Big](1 - \alpha_t^s) \\
&\quad + \mathbb{E}_{X_t|Y_t=1,S=s}\Big[ (1 - \pi_t^s(X_t))T_{10}^s + \pi_t^s(X_t)T_{11}^s \Big]\alpha_t^s \\
&= g^{0s}(\alpha_t^a, \alpha_t^b) \cdot (1 - \alpha_t^s) + g^{1s}(\alpha_t^a, \alpha_t^b) \cdot \alpha_t^s
\end{aligned}
$$

# F   Proofs

We define balanced equations and functions for the rest proofs. The dynamics system (4) can reach equilibrium if $\alpha_t^s = \alpha_{t-1}^s$ holds. Therefore, the system has equilibrium if there exists solution to the *balanced equations* defined as (5).

$$
\frac{1}{\alpha^a} - 1 = \frac{1 - g^{1a}(\theta^a(\alpha^a, \alpha^b))}{g^{0a}(\theta^a(\alpha^a, \alpha^b))}; \quad \frac{1}{\alpha^b} - 1 = \frac{1 - g^{1b}(\theta^b(\alpha^a, \alpha^b))}{g^{0b}(\theta^b(\alpha^a, \alpha^b))}. \tag{5}
$$

By removing subscript $t$ and writing threshold $\theta^s$ as a function of $\alpha^a, \alpha^b$, we have $g^{ys}(\theta^s(\alpha^a, \alpha^b)) = T_{y0}^s \mathbb{G}_y^s(\theta^s(\alpha^a, \alpha^b)) + T_{y1}^s(1 - \mathbb{G}_y^s(\theta^s(\alpha^a, \alpha^b)))$, denote CDF of $G_y^s(x)$ as $\mathbb{G}_y^s(\theta) = \int_{-\infty}^{\theta} G_y^s(x)dx$.

$\forall s \in \{a, b\}$, let $-s := \{a, b\} \setminus s$. $\forall \alpha^{-s} \in [0, 1]$, define *balanced set* w.r.t. dynamics as $\Psi^s(\alpha^{-s}) := \{\overline{\alpha}^s : \frac{1}{\overline{\alpha}^s} - 1 = \frac{1 - g^{1s}(\theta^a(\overline{\alpha}^s, \alpha^{-s}))}{g^{0s}(\theta^s(\overline{\alpha}^s, \alpha^{-s}))}\}$. If the set size $|\Psi^s(\alpha^{-s})| = 1$ holds $\forall \alpha^{-s} \in [0, 1]$, we define *balanced functions* w.r.t. dynamics as $\psi^s : [0, 1] \to [0, 1]$ with $\psi^s(\alpha^{-s}) \in \Psi^s(\alpha^{-s}), \forall \alpha^{-s} \in [0, 1]$.

**The proof that the threshold policies are optimal under our formulation.**

*Proof.* In the following proof, we focus on optimal policy at $t$ and omit the subscript $t$.

First consider unconstrained optimal policy, noted as $\pi_{\text{UN}}^s$, we have,

$$
\pi_{\text{UN}}^s = \arg\max_{\pi^s} \mathbb{E}_{X|S=s}[\pi^s(X)(\gamma^s(X)(u_+ + u_-) - u_-)]
$$

Therefore, the optimal policy satisfies $\pi_{\text{UN}}^s(x) = \mathbf{1}(\gamma^s(x) \geq \frac{u_-}{u_+ + u_-})$. Since $\gamma^s(x)$ is monotonically increasing in $x$ under Assumption 1, $\pi_{\text{UN}}^s(x) = \mathbf{1}(x \geq (\gamma^s)^{-1}(\frac{u_-}{u_+ + u_-}))$ is threshold policy where $(\gamma^s)^{-1}(\cdot)$ denotes the inverse function of $\gamma(\cdot)$.

Now consider optimal fair policy under some fairness constraint $\mathcal{C}$ satisfying Assumption 2. Consider any pair of policies $(\pi^a, \pi^b)$ that satisfies fairness constraint $\mathcal{C}$, and define fairness constant $c = \mathbb{E}_{X \sim \mathcal{P}_{\mathcal{C}}^a}[\pi^a(X)] = \mathbb{E}_{X \sim \mathcal{P}_{\mathcal{C}}^b}[\pi^b(X)] \in [0, 1]$. To show the optimal fair policy is threshold policy, we will show that there always exists a pair of threshold policies $(\pi_d^a, \pi_d^b)$ such that $\mathbb{E}_{X \sim \mathcal{P}_{\mathcal{C}}^a}[\pi_d^a(X)] = \mathbb{E}_{X \sim \mathcal{P}_{\mathcal{C}}^b}[\pi_d^b(X)] = c$, i.e., the fairness constant is the same as $(\pi^a, \pi^b)$, and the utility of $(\pi_d^a, \pi_d^b)$ is no less than the utility attained under $(\pi^a, \pi^b)$.

$\forall s \in \{a, b\}$, let threshold policy $\pi_d^s$ be defined such that $\pi_d^s(x) = \mathbf{1}(x \geq \theta_d^s)$ and $\mathbb{E}_{X \sim \mathcal{P}_\mathcal{C}^s}[\pi_d^s(X)] = c$ are satisfied. Such policy must exist and the threshold is given by $\theta_d^s = (\mathbb{P}_\mathcal{C}^s)^{-1}(1 - c)$, where $\mathbb{P}_\mathcal{C}^s(\theta^s) = \int_{-\infty}^{\theta^s} \mathcal{P}_\mathcal{C}^s(x) dx$ is CDF of $\mathcal{P}_\mathcal{C}^s$ and $(\mathbb{P}_\mathcal{C}^s)^{-1}(\cdot)$ is the inverse of it.

Let $R_{\pi_d^s}(D, Y)$, $R_{\pi^s}(D, Y)$ denote the utility attained under policies $\pi_d^s$, $\pi^s$ respectively. Next we will show that $\forall s \in \{a, b\}$, $\mathbb{E}[R_{\pi_d^s}(D, Y) \mid S = s] \geq \mathbb{E}[R_{\pi^s}(D, Y) \mid S = s]$ holds, i.e.,

$$\mathbb{E}_{X|S=s}[\pi_d^s(X)(\gamma^s(X)(u_+ + u_-) - u_-)] \geq \mathbb{E}_{X|S=s}[\pi^s(X)(\gamma^s(X)(u_+ + u_-) - u_-)]$$

Since $\pi_d^s(x) = \mathbf{1}(x \geq \theta_d^s)$, we have the followings,

$$\mathbb{E}_{X|S=s}[\pi_d^s(X)(\gamma^s(X)(u_+ + u_-) - u_-)] = \int_{\theta_d^s}^\infty (\gamma^s(x)(u_+ + u_-) - u_-)\mathbb{P}(X = x \mid S = s)dx$$

$$\mathbb{E}_{X|S=s}[\pi^s(X)(\gamma^s(X)(u_+ + u_-) - u_-)] = \int_{\theta_d^s}^\infty (\gamma^s(x)(u_+ + u_-) - u_-)\mathbb{P}(X = x \mid S = s)dx$$
$$+ \int_{-\infty}^{\theta_d^s} \pi^s(x)(\gamma^s(x)(u_+ + u_-) - u_-)\mathbb{P}(X = x \mid S = s)dx$$
$$- \int_{\theta_d^s}^\infty (1 - \pi^s(x))(\gamma^s(x)(u_+ + u_-) - u_-)\mathbb{P}(X = x \mid S = s)dx$$

Since $\mathbb{E}_{X \sim \mathcal{P}_\mathcal{C}^s}[\pi^s(X)] = c = \mathbb{E}_{X \sim \mathcal{P}_\mathcal{C}^s}[\pi_d^s(X)]$, we have

$$\int_{\theta_d^s}^\infty (1 - \pi^s(x))\mathcal{P}_\mathcal{C}^s(x)dx = \int_{-\infty}^{\theta_d^s} \pi^s(x)\mathcal{P}_\mathcal{C}^s(x)dx \tag{6}$$

Under Assumption 2, $\frac{\mathbb{P}(X=x|S=s)}{\mathcal{P}_\mathcal{C}^s(x)}$ is non-decreasing. Since $\gamma^s(x) = \alpha^s \frac{G_1^s(x)}{\mathbb{P}(X=x|S=s)}$ is non-decreasing and $1 - \gamma^s(x) = (1 - \alpha^s)\frac{G_0^s(x)}{\mathbb{P}(X=x|S=s)}$ is non-increasing, we have $\frac{G_1^s(x)}{\mathbb{P}(X=x|S=s)}$ is non-decreasing and $\frac{G_0^s(x)}{\mathbb{P}(X=x|S=s)}$ is non-increasing. Therefore,

$$(\gamma^s(x)(u_+ + u_-) - u_-)\frac{\mathbb{P}(X = x \mid S = s)}{\mathcal{P}_\mathcal{C}^s(x)}$$
$$= \alpha^s \frac{G_1^s(x)}{\mathcal{P}_\mathcal{C}^s(x)}u_+ - (1 - \alpha^s)\frac{G_0^s(x)}{\mathcal{P}_\mathcal{C}^s(x)}u_-$$
$$= \alpha^s \frac{G_1^s(x)}{\mathbb{P}(X = x \mid S = s)}\frac{\mathbb{P}(X = x \mid S = s)}{\mathcal{P}_\mathcal{C}^s(x)}u_+ - (1 - \alpha^s)\frac{G_0^s(x)}{\mathbb{P}(X = x \mid S = s)}\frac{\mathbb{P}(X = x \mid S = s)}{\mathcal{P}_\mathcal{C}^s(x)}u_-$$

is non-decreasing in $x$. Combine with Eqn. (6), we have the followings,

$$\int_{-\infty}^{\theta_d^s} \pi^s(x)(\gamma^s(x)(u_+ + u_-) - u_-)\mathbb{P}(X = x \mid S = s)dx$$
$$= \int_{-\infty}^{\theta_d^s} \pi^s(x)(\gamma^s(x)(u_+ + u_-) - u_-)\frac{\mathbb{P}(X = x \mid S = s)}{\mathcal{P}_\mathcal{C}^s(x)}\mathcal{P}_\mathcal{C}^s(x)dx$$
$$\leq \int_{-\infty}^{\theta_d^s} \pi^s(x)(\gamma^s(\theta_d^s)(u_+ + u_-) - u_-)\frac{\mathbb{P}(X = \theta_d^s \mid S = s)}{\mathcal{P}_\mathcal{C}^s(\theta_d^s)}\mathcal{P}_\mathcal{C}^s(x)dx$$
$$= \int_{\theta_d^s}^\infty (1 - \pi^s(x))(\gamma^s(\theta_d^s)(u_+ + u_-) - u_-)\frac{\mathbb{P}(X = \theta_d^s \mid S = s)}{\mathcal{P}_\mathcal{C}^s(\theta_d^s)}\mathcal{P}_\mathcal{C}^s(x)dx$$
$$\leq \int_{\theta_d^s}^\infty (1 - \pi^s(x))(\gamma^s(x)(u_+ + u_-) - u_-)\frac{\mathbb{P}(X = x \mid S = s)}{\mathcal{P}_\mathcal{C}^s(x)}\mathcal{P}_\mathcal{C}^s(x)dx$$
$$= \int_{\theta_d^s}^\infty (1 - \pi^s(x))(\gamma^s(x)(u_+ + u_-) - u_-)\mathbb{P}(X = x \mid S = s)dx.$$

Therefore, the following holds $\forall s \in \{a, b\}$,

$$\mathbb{E}_{X|S=s}[\pi_d^s(X)(\gamma^s(X)(u_+ + u_-) - u_-)] \geq \mathbb{E}_{X|S=s}[\pi^s(X)(\gamma^s(X)(u_+ + u_-) - u_-)].$$

It shows that the utility attained under threshold policy $(\pi_d^a, \pi_d^b)$ is no less than the utility of $(\pi^a, \pi^b)$, which concludes that the optimal fair policy $(\pi_\mathcal{C}^a, \pi_\mathcal{C}^b)$ must be threshold policies.

Lemma 2 below further shows that the optimal threshold policy $\theta^s(\alpha^a, \alpha^b)$ is continuous and non-increasing in $\alpha^a$ and $\alpha^b$.

**Lemma 2.** *Let* $\left(\theta^a(\alpha^a, \alpha^b), \theta^b(\alpha^a, \alpha^b)\right)$ *be a pair of solutions to Eqn.* (3) *under* $\alpha^a, \alpha^b$. $\forall s \in \{a, b\}$, *if* $\frac{G_1^s(x)}{\mathcal{P}_{\mathcal{C}}^s(x)}$ *and* $\frac{G_0^s(x)}{\mathcal{P}_{\mathcal{C}}^s(x)}$ *are continuous everywhere in* $x$, *then* $\theta^s(\alpha^a, \alpha^b)$ *is continuous in both* $\alpha^a$ *and* $\alpha^b$. *Moreover, under Assumption 2,* $\theta^s(\alpha^a, \alpha^b)$ *is non-increasing in* $\alpha^a$ *and* $\alpha^b$.

*Proof.* To prove that *a sufficient condition under which* $\theta^s(\alpha^a, \alpha^b)$ *is continuous in* $\alpha^a, \alpha^b \in [0, 1]$ *is that* $\frac{G_1^s(x)}{\mathcal{P}_{\mathcal{C}}^s(x)}$ *and* $\frac{G_0^s(x)}{\mathcal{P}_{\mathcal{C}}^s(x)}$ *are continuous everywhere in* $x$, we define a function $f^s(\theta^s, \alpha^a, \alpha^b)$:

$$
\begin{aligned}
f^s(\theta^s, \alpha^a, \alpha^b) &= (\gamma^s(\theta^s) - \frac{u_-}{u_+ + u_-})\frac{\mathbb{P}(X = \theta^s \mid S = s)}{\mathcal{P}_{\mathcal{C}}^s(\theta^s)} \\
&= [\alpha^s u_+ G_1^s(\theta^s) + \alpha^s u_- G_0^s(\theta^s) - u_- G_0^s(\theta^s)]\frac{1}{\mathcal{P}_{\mathcal{C}}^s(\theta^s)} \\
&= [\alpha^s \frac{G_1^s(\theta^s)}{\mathcal{P}_{\mathcal{C}}^s(\theta^s)}u_+ + (\alpha^s - 1)\frac{G_0^s(\theta^s)}{\mathcal{P}_{\mathcal{C}}^s(\theta^s)}u_-].
\end{aligned}
$$

According to Equation (3), we have $p_a f^a(\theta^a, \alpha^a, \alpha^b) + p_b f^b(\theta^b, \alpha^a, \alpha^b) = 0$.

Given any $\alpha^a$ and $\alpha^b$, and any constant $k$, let $\tilde{\theta}_i^s$ be one solution to $f^s(\theta^s, \alpha^a, \alpha^b) = k$, where $i = 1, ..., N$ and $N$ is the number of solutions. Firstly, we show that $\tilde{\theta}_i^s(\alpha^a, \alpha^b)$ is continuous in $\alpha^a$ and $\alpha^b$, for any $i \in \{1, ..., N\}$. Because $\frac{G_1^s(x)}{\mathcal{P}_{\mathcal{C}}^s(x)}$ and $\frac{G_0^s(x)}{\mathcal{P}_{\mathcal{C}}^s(x)}$ are continuous, $f^s(\theta^s, \alpha^a, \alpha^b)$ is continuous in $\alpha^a$, $\alpha^b$, and $\theta^s$. Therefore, $\forall \epsilon > 0, \exists \delta > 0$ such that for all $|\alpha^{a'} - \alpha^a| < \delta$ and $|\alpha^{b'} - \alpha^b| < \delta \implies |\tilde{\theta}_i^{s'} - \tilde{\theta}_i^s| < \epsilon$. Thus, $\tilde{\theta}_i^s(\alpha^a, \alpha^b)$ is continuous in $\alpha^a$ and $\alpha^b$, $\forall i \in \{1, ..., N\}$.

Next, we show that given $\alpha^a$ and $\alpha^b$, the solutions to $p_a f^a(\theta^a, \alpha^a, \alpha^b) + p_b f^b(\theta^b, \alpha^a, \alpha^b) = 0$ under fairness constraint $\mathcal{C}$ are continuous in $\alpha^a$ and $\alpha^b \in [0, 1]$. Under fairness constraints in Equation (1), $\theta^a = \phi_{\mathcal{C}}(\theta^b)$ holds for some continuous function $\phi_{\mathcal{C}}(\cdot)$. Consequently, we have $p_a f^a(\phi_{\mathcal{C}}(\theta^b), \alpha^a, \alpha^b) + p_b f^b(\theta^b, \alpha^a, \alpha^b) = 0$. Because $f^s(\cdot, \cdot, \cdot)$ and $\phi_{\mathcal{C}}(\cdot)$ are continuous functions, with the same reasoning, given $\alpha^a$ and $\alpha^b$, the solutions to $p_a f^a(\phi_{\mathcal{C}}(\theta^b), \alpha^a, \alpha^b) + p_b f^b(\theta^b, \alpha^a, \alpha^b) = 0$ are continuous in $\alpha^a$ and $\alpha^b$. In other words, $\theta_i^s(\alpha^a, \alpha^b)$ is continuous.

Under Assumption 2, $f^s(\theta^s, \alpha^a, \alpha^b)$ and $\theta^s(\alpha^a, \alpha^b)$ are continuous. We then prove that if $\frac{G_1^s(x)}{\mathcal{P}_{\mathcal{C}}^s(x)}$ is non-decreasing and $\frac{G_0^s(x)}{\mathcal{P}_{\mathcal{C}}^s(x)}$ is non-increasing in $x$, then $\theta^s(\alpha^a, \alpha^b)$ is non-increasing in $\alpha^a$ and $\alpha^b$.

Let $(\phi_{\mathcal{C}}(\theta^b), \theta^b)$ be a pair that satisfies fairness constraint, where $\phi_{\mathcal{C}}(\cdot)$ is some continuous and strictly increasing function, then the optimal one is the pair that satisfies Equation (3) as follows:

$$
p_a(\gamma^a(\phi_{\mathcal{C}}(\theta^b)) - \frac{u_-}{u_+ + u_-})\frac{\mathbb{P}(X = \phi_{\mathcal{C}}(\theta^b)|S = a)}{\mathcal{P}_{\mathcal{C}}^a(\phi_{\mathcal{C}}(\theta^b))} + p_b(\gamma^b(\theta^b) - \frac{u_-}{u_+ + u_-})\frac{\mathbb{P}(X = \theta^b|S = b)}{\mathcal{P}_{\mathcal{C}}^b(\theta^b)}
$$
$$
= p_a\left[\alpha^a \frac{G_1^a(\phi_{\mathcal{C}}(\theta^b))}{\mathcal{P}_{\mathcal{C}}^a(\phi_{\mathcal{C}}(\theta^b))}u_+ + (\alpha^a - 1)\frac{G_0^a(\phi_{\mathcal{C}}(\theta^b))}{\mathcal{P}_{\mathcal{C}}^a(\phi_{\mathcal{C}}(\theta^b))}u_-\right] + p_b\left[\alpha^b \frac{G_1^b(\theta^b)}{\mathcal{P}_{\mathcal{C}}^b(\theta^b)}u_+ + (\alpha^b - 1)\frac{G_0^b(\theta^b)}{\mathcal{P}_{\mathcal{C}}^b(\theta^b)}u_-\right] = 0.
$$

Note that $\forall s \in \{a, b\}$, LHS of above equation is strictly increasing in $\alpha^s$ because the coefficient of $\alpha^s$ is positive. Because $\frac{G_1^s(x)}{\mathcal{P}_{\mathcal{C}}^s(x)}$ is non-decreasing and $\frac{G_0^s(x)}{\mathcal{P}_{\mathcal{C}}^s(x)}$ is non-increasing in $x$, $\frac{G_1^s(x)}{\mathcal{P}_{\mathcal{C}}^s(x)} - \frac{G_0^s(x)}{\mathcal{P}_{\mathcal{C}}^s(x)}$ is non-decreasing in $x$. As $\alpha^s$ increases, both $\frac{G_1^a(\phi_{\mathcal{C}}(\theta^b))}{\mathcal{P}_{\mathcal{C}}^a(\phi_{\mathcal{C}}(\theta^b))} - \frac{G_0^a(\phi_{\mathcal{C}}(\theta^b))}{\mathcal{P}_{\mathcal{C}}^a(\phi_{\mathcal{C}}(\theta^b))}$ and $\frac{G_1^b(\theta^b)}{\mathcal{P}_{\mathcal{C}}^b(\theta^b)} - \frac{G_0^b(\theta^b)}{\mathcal{P}_{\mathcal{C}}^b(\theta^b)}$ must not increase so that the optimal fair equation can be maintained. It requires that both $\theta^b$ and $\theta^a = \phi_{\mathcal{C}}(\theta^b)$ must not increase. In other words, $\forall s \in \{a, b\}$, $\theta^s(\alpha^a, \alpha^b)$ must be non-increasing in $\alpha^a$ and $\alpha^b$. $\square$

$\square$

**The proof of Lemma 1.**

*Proof.* In the following proof, we focus on optimal policy at $t$ and omit the subscript $t$.

First consider unconstrained optimal policy. Under threshold policy,

$$\theta_{\text{UN}}^{s*} \;=\; \arg\max_{\theta^s} \mathbb{E}_{X|S=s}[\pi^s(X)(\gamma^s(X)(u_+ + u_-) - u_-)]$$

$$=\; \arg\max_{\theta^s} \int_{\theta^s}^{\infty} (\gamma^s(x)(u_+ + u_-) - u_-)\mathbb{P}(X = x \mid S = s)dx$$

Since $\gamma^s(x)$ is monotonically increasing in $x$ under Assumption 1, $\theta_{\text{UN}}^{s*}$ satisfies $\gamma^s(\theta_{\text{UN}}^{s*}) = \frac{u_-}{u_+ + u_-}$.

Now consider optimal policy under fairness constraint, to satisfy constraint $\mathcal{C}$, $\int_{\theta^a}^{\infty} \mathcal{P}_{\mathcal{C}}^a(x)dx = \int_{\theta^b}^{\infty} \mathcal{P}_{\mathcal{C}}^b(x)dx$ should hold. Denote CDF $\mathbb{P}_{\mathcal{C}}^s(\theta^s) = \int_{-\infty}^{\theta^s} \mathcal{P}_{\mathcal{C}}^s(x)dx$, then for any pair $(\theta^a, \theta^b)$ that is fair, we have $\theta^a = (\mathbb{P}_{\mathcal{C}}^a)^{-1}\mathbb{P}_{\mathcal{C}}^b(\theta^b) = \phi_{\mathcal{C}}(\theta^b)$ hold for some strictly increasing function $\phi_{\mathcal{C}}(\cdot)$. Denote $u = \mathbb{P}_{\mathcal{C}}^b(\theta^b)$ and $\theta^a = (\mathbb{P}_{\mathcal{C}}^a)^{-1}(u)$, the following holds,

$$\frac{d\phi_{\mathcal{C}}(\theta^b)}{d\theta^b} = \frac{d(\mathbb{P}_{\mathcal{C}}^a)^{-1}\mathbb{P}_{\mathcal{C}}^b(\theta^b)}{d\theta^b} = \frac{d(\mathbb{P}_{\mathcal{C}}^a)^{-1}(u)}{du}\frac{du}{d\theta^b} = \frac{1}{(\mathbb{P}_{\mathcal{C}}^a)'((\mathbb{P}_{\mathcal{C}}^a)^{-1}(u))}\frac{du}{d\theta^b} = \frac{(\mathbb{P}_{\mathcal{C}}^b)'(\theta^b)}{(\mathbb{P}_{\mathcal{C}}^a)'(\theta^a)} = \frac{\mathcal{P}_{\mathcal{C}}^b(\theta^b)}{\mathcal{P}_{\mathcal{C}}^a(\theta^a)}.$$

Denote $f^s(x) := (\gamma^s(x)(u_+ + u_-) - u_-)\mathbb{P}(X = x \mid S = s)$, then we have

$$\theta_{\mathcal{C}}^{b*} = \arg\max_{\theta^b}\mathcal{U}(D, Y) = \arg\max_{\theta^b}\left( p_a \int_{\phi_{\mathcal{C}}(\theta^b)}^{\infty} f^a(x)dx + p_b \int_{\theta^b}^{\infty} f^b(x)dx \right).$$

Let $F(\theta^b) := p_a \int_{\phi_{\mathcal{C}}(\theta^b)}^{\infty} f^a(x)dx + p_b \int_{\theta^b}^{\infty} f^b(x)dx$. Because $\gamma^s(x)$ is monotonically increasing in $x$ under Assumption 1, the optimal $\theta_{\mathcal{C}}^{b*}$ satisfies

$$\frac{dF(\theta^b)}{d\theta^b}\bigg|_{\theta^b = \theta_{\mathcal{C}}^{b*}} \;=\; -p_a f^a(\phi_{\mathcal{C}}(\theta^b))\frac{d\phi_{\mathcal{C}}(\theta^b)}{d\theta^b} - p_b f^b(\theta^b)\bigg|_{\theta^b = \theta_{\mathcal{C}}^{b*}}$$

$$=\; -p_a(\gamma^a(\phi_{\mathcal{C}}(\theta_{\mathcal{C}}^{b*}))(u_+ + u_-) - u_-)\mathbb{P}(X = \phi_{\mathcal{C}}(\theta_{\mathcal{C}}^{b*}) \mid S = a)\frac{\mathcal{P}_{\mathcal{C}}^b(\theta_{\mathcal{C}}^{b*})}{\mathcal{P}_{\mathcal{C}}^a(\phi_{\mathcal{C}}(\theta_{\mathcal{C}}^{b*}))}$$

$$\quad -p_b(\gamma^b(\theta_{\mathcal{C}}^{b*})(u_+ + u_-) - u_-)\mathbb{P}(X = \theta_{\mathcal{C}}^{b*} \mid S = b)$$

$$=\; 0.$$

Therefore,

$$p_a(\gamma^a(\theta_{\mathcal{C}}^{a*})(u_+ + u_-) - u_-)\frac{\mathbb{P}(X=\theta_{\mathcal{C}}^{a*}|S=a)}{\mathcal{P}_{\mathcal{C}}(\theta_{\mathcal{C}}^{a*})} + p_b(\gamma^b(\theta_{\mathcal{C}}^{b*})(u_+ + u_-) - u_-)\frac{\mathbb{P}(X=\theta_{\mathcal{C}}^{b*}|S=b)}{\mathcal{P}_{\mathcal{C}}(\theta_{\mathcal{C}}^{b*})} = 0.$$

$\square$

**The proof of Theorem 1.**

*Proof.* $\forall s \in \{a, b\}$, define function $l^s(\alpha^s) := \frac{1}{\alpha^s} - 1$ and $h^s(\theta^s(\alpha^a, \alpha^b)) := \frac{1 - g^{1s}(\theta^s(\alpha^a, \alpha^b))}{g^{0s}(\theta^s(\alpha^a, \alpha^b))}$,

$$h^s(\theta^s(\alpha^a, \alpha^b)) = \frac{1 - (T_{10}^s\mathbb{G}_1^s(\theta^s(\alpha^a, \alpha^b)) + T_{11}^s(1 - \mathbb{G}_1^s(\theta^s(\alpha^a, \alpha^b))))}{T_{00}^s\mathbb{G}_0^s(\theta^s(\alpha^a, \alpha^b)) + T_{01}^s(1 - \mathbb{G}_0^s(\theta^s(\alpha^a, \alpha^b)))}.$$

Firstly, we prove that given a fixed $\alpha^{-s} \in [0, 1]$ there must exist at least one $\overline{\alpha}^s \in (0, 1)$ such that $h^s(\theta^s(\alpha^{-s}, \overline{\alpha}^s)) = l^s(\overline{\alpha}^s)$, $s \in \{a, b\}$, $-s = \{a, b\} \setminus s$.

Since $\mathbb{G}_y^s(x)$ is continuous in $x$, and $\theta^s(\alpha^a, \alpha^b)$ is continuous in $\alpha^a$ and $\alpha^b$, $\mathbb{G}_y^s(\theta^s(\alpha^a, \alpha^b))$ is continuous in $\alpha^a$ and $\alpha^b$. Therefore, $h^s(\theta^s(\alpha^a, \alpha^b))$ is continuous in $\alpha^a$ and $\alpha^b$.

Moreover, $g^{1s}(\theta^s(\alpha^a, \alpha^b))$ is the convex combination of $T_{11}^s$ and $T_{10}^s$, and $g^{0s}(\theta^s(\alpha^a, \alpha^b))$ is the convex combination of $T_{01}^s$ and $T_{00}^s$, the following holds $\forall \alpha^a \in [0, 1], \alpha^b \in [0, 1]$,

$$\min\{T_{10}^s, T_{11}^s\} \le g^{1s}(\theta^s(\alpha^a, \alpha^b)) \le \max\{T_{10}^s, T_{11}^s\};$$
$$\min\{T_{00}^s, T_{01}^s\} \le g^{0s}(\theta^s(\alpha^a, \alpha^b)) \le \max\{T_{00}^s, T_{01}^s\},$$

which implies $0 < \frac{1 - \max\{T_{10}^s, T_{11}^s\}}{\max\{T_{00}^s, T_{01}^s\}} \le h^s(\theta^s(\alpha^a, \alpha^b)) \le \frac{1 - \min\{T_{10}^s, T_{11}^s\}}{\min\{T_{00}^s, T_{01}^s\}} < +\infty$.

Furthermore, $l^s(\alpha^s) := \frac{1}{\alpha^s} - 1$ is continuous and strictly decreasing in $\alpha^s$, and

$$\lim_{\alpha^s \to 0} l^s(\alpha^s) = +\infty; \quad \lim_{\alpha^s \to 1} l^s(\alpha^s) = 0,$$

Given a fixed $\alpha^a \in [0, 1]$, because $h^b(\theta^b(\alpha^a, \alpha^b))$ is continuous over $\alpha^b \in [0, 1]$ and with value varying between $\frac{1-\max\{T_{10}^b, T_{11}^b\}}{\max\{T_{00}^b, T_{01}^b\}}$ and $\frac{1-\min\{T_{10}^b, T_{11}^b\}}{\min\{T_{00}^b, T_{01}^b\}}$, and $l^b(\alpha^b)$ is continuous with value varying from $+\infty$ to 0, there must exist at least one $\overline{\alpha}^b \in (0, 1)$ such that $h^b(\theta^b(\alpha^a, \overline{\alpha}^b)) = l^b(\overline{\alpha}^b)$. Similarly, given a fixed $\alpha^b \in [0, 1]$, there must exist at least one $\overline{\alpha}^a \in (0, 1)$ such that $h^a(\theta^a(\overline{\alpha}^a, \alpha^b)) = l^a(\overline{\alpha}^a)$.

Secondly, we prove that all the solutions $(\overline{\alpha}^a, \alpha^b)$ and $(\alpha^a, \overline{\alpha}^b)$ are on continuous curves in the 2D plane $\{(\alpha^a, \alpha^b) : \alpha^a \in [0, 1], \alpha^b \in [0, 1]\}$.

According to the continuity of $l^s(\cdot)$ and $h^s(\cdot)$, we have $\forall \alpha^a \in [0, 1], \lim_{\alpha^{a'} \to \alpha^a} l^a(\alpha^{a'}) = l^a(\alpha^a)$; furthermore, $\forall \alpha^a \in [0, 1]$ and $\forall \theta_i^a \in \{\theta^a : l^a(\alpha^a) = h^a(\theta^a)\}, \lim_{\theta_i^{a'} \to \theta_i^a} h^a(\theta_i^{a'}) = h^a(\theta_i^a)$. Thus, $\forall \epsilon > 0, \exists \delta > 0$, such that $\forall \alpha^a \in [0, 1], |\alpha^{a'} - \alpha^a| < \delta \Longrightarrow |\theta_i^{a'} - \theta_i^a| < \epsilon$. Consequently, $\forall \epsilon > 0, \exists \delta' > 0$ and $\exists \delta > 0$, such that $\forall \alpha^a \in [0, 1], |\alpha^{a'} - \alpha^a| < \delta \Longrightarrow |\theta_i^{a'} - \theta_i^a| < \delta' \Longrightarrow |\alpha_i^{b'} - \alpha_i^b| < \epsilon$, the last statement is because of the continuity of $\theta^a(\alpha^a, \alpha^b)$; in other words, $\forall \alpha^a \in [0, 1], \lim_{\alpha^{a'} \to \alpha^a} \alpha_i^{b'} = \alpha_i^b$, where $i = 1, ..., N$. Therefore, $(\overline{\alpha}^a, \alpha^b)$ is on a set of continuous curves with $\alpha^b$ varying from 0 to 1. Similarly, one can prove that $(\alpha^a, \overline{\alpha}^b)$ is also on a set of continuous curves with $\alpha^a$ varying from 0 to 1.

Finally, we show the existence of equilibrium $(\widehat{\alpha}^a, \widehat{\alpha}^b)$.

Consider a 2D plane $\{(\alpha^a, \alpha^b) : \alpha^a \in [0, 1], \alpha^b \in [0, 1]\}$, and $\mathcal{C}_1 = \{(\overline{\alpha}^a, \alpha^b)\}$ and $\mathcal{C}_2 = \{(\alpha^a, \overline{\alpha}^b)\}$ that are two sets of continuous curves in the plane defined earlier. It is straightforward to see that there is at least one curve among $\mathcal{C}_1$ whose $\alpha^b$ varies from 0 to 1 and at least one curve among $\mathcal{C}_2$ whose $\alpha^a$ varies from 0 to 1. These two continuous curves must have at least one intersection. Moreover, this intersection $(\widehat{\alpha}^a, \widehat{\alpha}^b)$ satisfies $h^b(\theta^b(\widehat{\alpha}^a, \widehat{\alpha}^b)) = l^b(\widehat{\alpha}^b)$ and $h^a(\theta^a(\widehat{\alpha}^a, \widehat{\alpha}^b)) = l^a(\widehat{\alpha}^a)$, is an equilibrium of system.

Moreover, we also realized that the proof can also be done by using Brouwer's Fixed Point Theorem in topology. $\qquad \square$

**The proof of Theorem 2.**

*Proof.* Following the proof of Theorem 1,

$$h^s(\theta^s(\alpha^a, \alpha^b)) = \frac{1 - g^{1s}(\theta^s(\alpha^a, \alpha^b))}{g^{0s}(\theta^s(\alpha^a, \alpha^b))} = \frac{1 - (T_{10}^s \mathbb{G}_1^s(\theta^s(\alpha^a, \alpha^b)) + T_{11}^s (1 - \mathbb{G}_1^s(\theta^s(\alpha^a, \alpha^b))))}{T_{00}^s \mathbb{G}_0^s(\theta^s(\alpha^a, \alpha^b)) + T_{01}^s (1 - \mathbb{G}_0^s(\theta^s(\alpha^a, \alpha^b)))}.$$

Note that $\forall y \in \{0, 1\}, T_{y0}^s \mathbb{G}_y^s(\theta^s(\alpha^a, \alpha^b)) + T_{y1}^s (1 - \mathbb{G}_y^s(\theta^s(\alpha^a, \alpha^b)))$ is the convex combination of $T_{y0}^s$ and $T_{y1}^s$ with CDF $\mathbb{G}_y^s(\theta^s(\alpha^a, \alpha^b))$ as the weight. Because $\mathbb{G}_y^s(\theta^s(\alpha^a, \alpha^b))$ is continuous and non-decreasing in $\theta^s(\alpha^a, \alpha^b)$, under Condition 1(A), $h^s(\theta^s(\alpha^a, \alpha^b))$ is non-decreasing in $\theta^s(\alpha^a, \alpha^b)$; while under Condition 1(B), $h^s(\theta^s(\alpha^a, \alpha^b))$ is non-increasing in $\theta^s(\alpha^a, \alpha^b)$.

Under unconstrained optimal policy or optimal fair policy with constraint $\mathcal{C}$ satisfying Assumption 1 and 2, $\theta^s(\alpha^a, \alpha^b)$ is non-increasing in $\alpha^a, \alpha^b$. Therefore, under Condition 1(A), $h^s(\theta^s(\alpha^a, \alpha^b))$ is non-decreasing in $\alpha^a, \alpha^b$, while under Condition 1(B), $h^s(\theta^s(\alpha^a, \alpha^b))$ is non-increasing in $\alpha^a, \alpha^b$. Moreover,

$$\text{Under Condition 1(A):} \quad 0 < \frac{1 - T_{10}^s}{T_{00}^s} \leq h^s(\theta^s(\alpha^a, \alpha^b)) \leq \frac{1 - T_{11}^s}{T_{01}^s} < +\infty$$

$$\text{Under Condition 1(B):} \quad 0 < \frac{1 - T_{11}^s}{T_{01}^s} \leq h^s(\theta^s(\alpha^a, \alpha^b)) \leq \frac{1 - T_{10}^s}{T_{00}^s} < +\infty$$

First consider the case when Condition 1(A) is satisfied.

Because function $l^s(\alpha^s) := \frac{1}{\alpha^s} - 1$ is continuous and strictly decreasing from $+\infty$ to 0 over $\alpha^s \in [0, 1], \forall s \in \{a, b\}$. Thus, given any fixed $\alpha^b \in [0, 1]$, strictly decreasing function $l^a(\alpha^a)$ and

non-decreasing function $h^a(\theta^a(\alpha^a, \alpha^b))$ has exactly one intersection, i.e., $\exists$ only one $\overline{\alpha}^a$ such that $h^a(\theta^a(\overline{\alpha}^a, \alpha^b)) = l^a(\overline{\alpha}^a)$. $\forall \alpha^b$, the set $\Psi^a(\alpha^b) = \{\overline{\alpha}^a : h^a(\theta^a(\overline{\alpha}^a, \alpha^b)) = l^a(\overline{\alpha}^a)\}$ has only one element, and they constitute continuous function $\overline{\alpha}^a = \psi^a(\alpha^b)$ (balanced function). Similarly, $\forall \alpha^a$, set $\Psi^b(\alpha^a) = \{\overline{\alpha}^b : h^b(\theta^b(\alpha^a, \overline{\alpha}^b)) = l^b(\overline{\alpha}^b)\}$ also has only one element, which forms continuous function $\overline{\alpha}^b = \psi^b(\alpha^a)$.

Because given any $\alpha^a$, $h^a(\theta^a(\alpha^a, \alpha^b))$ is non-decreasing in $\alpha^b$, as $\alpha^b$ increases, the intersection with $l^a(\overline{\alpha}^a)$ is non-increasing. Therefore, $\psi^a(\alpha^b)$ is non-increasing in $\alpha^b$. Similarly, $\psi^b(\alpha^a)$ is also non-increasing in $\alpha^a$.

On the 2D plane $\{(\alpha^a, \alpha^b) : \alpha^a \in [0,1], \alpha^b \in [0,1]\}$, two curves $\mathcal{C}_1 = \{(\alpha^a, \alpha^b) : \alpha^a = \psi^a(\alpha^b), \alpha^b \in [0,1]\}$ and $\mathcal{C}_2 = \{(\alpha^a, \alpha^b) : \alpha^b = \psi^b(\alpha^a), \alpha^a \in [0,1]\}$ are both continuous and non-increasing. One sufficient condition to guarantee $\mathcal{C}_1$ and $\mathcal{C}_2$ have exact one intersection, is that $|\frac{d\psi^a(\alpha^b)}{d\alpha^b}| < 1, \forall \alpha^b \in [0,1]$ and $|\frac{d\psi^b(\alpha^a)}{d\alpha^a}| < 1, \forall \alpha^a \in [0,1]$. In the followings, we show these sufficient conditions will hold if $|\frac{\partial h^a(\theta^a(\alpha^a, \alpha^b))}{\partial \alpha^b}| < 1$ and $|\frac{\partial h^b(\theta^b(\alpha^a, \alpha^b))}{\partial \alpha^a}| < 1, \forall \alpha^a, \alpha^b$.

Denote $u := h^a(\theta^a(\psi^a(\alpha^b), \alpha^b))$, because $l^a(\psi^a(\alpha^b)) = h^a(\theta^a(\psi^a(\alpha^b), \alpha^b)), \forall \alpha^b$,

$$\frac{d\psi^a(\alpha^b)}{d\alpha^b} = \frac{d(l^a)^{-1}(u)}{d\alpha^b} = \frac{d(l^a)^{-1}(u)}{du}\frac{du}{d\alpha^b} = \frac{1}{(l^a)'((l^a)^{-1}(u))}\frac{du}{d\alpha^b} = -((l^a)^{-1}(u))^2\frac{du}{d\alpha^b}.$$

Because $(l^a)^{-1}(u) = \psi^a(\alpha^b) \in [0,1]$, $-((l^a)^{-1}(u))^2 \in [-1, 0]$. Moreover, because of the condition $|\frac{dh^a(\theta^a(\alpha^a, \alpha^b))}{d\alpha^b}| < 1$, we have

$$\left|\frac{d\psi^a(\alpha^b)}{d\alpha^b}\right| < 1.$$

Similarly, we can show that $|\frac{d\psi^b(\alpha^a)}{d\alpha^a}| < 1$ holds $\forall \alpha^a$ if $|\frac{\partial h^b(\theta^b(\alpha^a, \alpha^b))}{\partial \alpha^a}| < 1$. Therefore, $\mathcal{C}_1, \mathcal{C}_2$ have only one intersection, the equilibrium $(\widehat{\alpha}^a, \widehat{\alpha}^b)$ is unique.

Now consider the case when Condition 1(B) is satisfied.

Because $\frac{dl^s(\alpha^s)}{d\alpha^s} = -\frac{1}{(\alpha^s)^2} < -1, \forall \alpha^s \in (0,1)$, and $-1 \leq \frac{\partial h^s(\theta^s(\alpha^a, \alpha^b))}{\partial \alpha^s} \leq 0$ for any fixed $\alpha^{-s} \in [0,1]$. Strictly decreasing function $l^s(\alpha^s)$ and non-increasing function $h^s(\theta^s(\alpha^a, \alpha^b))$ has exactly one intersection. Therefore, $\forall \alpha^b$, balanced set $\Psi^a(\alpha^b) = \{\overline{\alpha}^a : h^a(\theta^a(\overline{\alpha}^a, \alpha^b)) = l^a(\overline{\alpha}^a)\}$ has only one element, and they constitute continuous function $\overline{\alpha}^a = \psi^a(\alpha^b)$ (balanced function). Similarly, $\forall \alpha^b$, set $\Psi^a(\alpha^b) = \{\overline{\alpha}^a : h^a(\theta^a(\overline{\alpha}^a, \alpha^b)) = l^a(\overline{\alpha}^a)\}$ also has only one element, which forms continuous function $\overline{\alpha}^a = \psi^a(\alpha^b)$.

Because given any $\alpha^a$, $h^a(\theta^a(\alpha^a, \alpha^b))$ is non-increasing in $\alpha^b$. As $\alpha^b$ increases, the intersection with $l^a(\overline{\alpha}^a)$ is non-decreasing. Therefore, $\psi^a(\alpha^b)$ is non-decreasing in $\alpha^b$. Similarly, $\psi^b(\alpha^a)$ is also non-decreasing in $\alpha^a$.

On the 2D plane $\{(\alpha^a, \alpha^b) : \alpha^a \in [0,1], \alpha^b \in [0,1]\}$, two curves $\mathcal{C}_1 = \{(\alpha^a, \alpha^b) : \alpha^a = \psi^a(\alpha^b), \alpha^b \in [0,1]\}$ and $\mathcal{C}_2 = \{(\alpha^a, \alpha^b) : \alpha^b = \psi^b(\alpha^a), \alpha^a \in [0,1]\}$ are both continuous and non-decreasing. One sufficient condition to guarantee $\mathcal{C}_1$ and $\mathcal{C}_2$ have exact one intersection, is that $\frac{d\psi^a(\alpha^b)}{d\alpha^b} < 1, \forall \alpha^b \in [0,1]$ and $\frac{d\psi^b(\alpha^a)}{d\alpha^a} < 1, \forall \alpha^a \in [0,1]$. Using the same analysis as the case under Condition 1(A), we can show these sufficient conditions will hold if $|\frac{\partial h^a(\theta^a(\alpha^a, \alpha^b))}{\partial \alpha^b}| < 1$ and $|\frac{\partial h^b(\theta^b(\alpha^a, \alpha^b))}{\partial \alpha^a}| < 1, \forall \alpha^a, \alpha^b$.

Therefore, $\mathcal{C}_1, \mathcal{C}_2$ have only one intersection, the equilibrium $(\widehat{\alpha}^a, \widehat{\alpha}^b)$ is unique. $\qquad\square$

**The proof of Corollary 1.**

**Corollary 1.** *For any feature distribution $\{G_y^s(x)\}_{s,y}$, suppose that $\left|\frac{\partial \mathbb{G}_y^s(\theta^s(\alpha^a, \alpha^b))}{\partial \alpha^u}\right| \leq M_y$ holds for some constant $M_y \in [0, \infty)$, $\forall y \in \{0,1\}, \forall u \in \{a, b\}$. Under either Condition 1(A) or 1(B), $\exists \epsilon_y^s > 0$ such that for any transitions that satisfy $|T_{y1}^s - T_{y0}^s| < \epsilon_y^s$, $s \in \{a, b\}, y \in \{0, 1\}$, the corresponding dynamics system has a unique equilibrium.*

*Proof.* Define notations $\mathbb{G}_y^s := \mathbb{G}_y^s(\theta^s(\alpha^a, \alpha^b))$, $\Delta T_0^s := T_{01}^s - T_{00}^s$ and $\Delta T_1^s := T_{11}^s - T_{10}^s$.

$$h^s(\theta^s(\alpha^a, \alpha^b)) = \frac{(1 - T_{10}^s)\mathbb{G}_1^s + (1 - T_{11}^s)(1 - \mathbb{G}_1^s)}{T_{00}^s\mathbb{G}_0^s + T_{01}^s(1 - \mathbb{G}_0^s)} = \frac{(1 - T_{11}^s) + \Delta T_1^s\mathbb{G}_1^s}{T_{00}^s + \Delta T_0^s(1 - \mathbb{G}_0^s)}$$

Take derivative w.r.t. $\alpha^u$, $\forall u \in \{a, b\}$,

$$\frac{\partial h^s(\theta^s(\alpha^a, \alpha^b))}{\partial \alpha^u} = \frac{\Delta T_1^s\frac{\partial\mathbb{G}_1^s}{\partial\alpha^u}(T_{00}^s + \Delta T_0^s(1 - \mathbb{G}_0^s)) + \Delta T_0^s\frac{\partial\mathbb{G}_0^s}{\partial\alpha^u}((1 - T_{11}^s) + \Delta T_1^s\mathbb{G}_1^s)}{(T_{00}^s + \Delta T_0^s(1 - \mathbb{G}_0^s))^2}$$

Consider case under Condition 1(A). Since $\Delta T_0^s < 0$, $\Delta T_1^s < 0$, $T_{00}^s + \Delta T_0^s(1 - \mathbb{G}_0^s) > 0$, and $(1 - T_{11}^s) + \Delta T_1^s\mathbb{G}_1^s > 0$, we have $|\frac{\partial h^s(\theta^s(\alpha^a, \alpha^b))}{\partial\alpha^u}| \leq |\frac{\Delta T_1^s M_1 T_{00}^s + \Delta T_0^s M_0(1 - T_{11}^s)}{(T_{01}^s)^2}|$.

Take $\epsilon_1^s = \epsilon_0^s = \frac{(T_{01}^s)^2}{M_1 T_{00}^s + M_0(1 - T_{11}^s)}$, if $|\Delta T_1^s| < \epsilon_1^s$ and $|\Delta T_0^s| < \epsilon_0^s$, then $|\frac{\partial h^s(\theta^s(\alpha^a, \alpha^b))}{\partial\alpha^u}| < 1$ holds. From Theorem 2, the equilibrium of dynamics 4 is unique.

Consider case under Condition 1(B).

Since $\Delta T_0^s > 0$ and $\Delta T_1^s > 0$, we have $|\frac{\partial h^s(\theta^s(\alpha^a, \alpha^b))}{\partial\alpha^u}| \leq \frac{\Delta T_1^s M_1 T_{01}^s + \Delta T_0^s M_0(1 - T_{10}^s)}{(T_{00}^s)^2}$.

Take $\epsilon_1^s = \epsilon_0^s = \frac{(T_{00}^s)^2}{M_1 T_{01}^s + M_0(1 - T_{10}^s)}$, if $\Delta T_1^s < \epsilon_1^s$ and $\Delta T_0^s < \epsilon_0^s$, then $|\frac{\partial h^s(\theta^s(\alpha^a, \alpha^b))}{\partial\alpha^u}| < 1$ holds. From Theorem 2, the equilibrium of dynamics 4 is unique. $\square$

**The proof of Theorem 3.**

*Proof.* $\forall s \in \{a, b\}$, an equilibrium $\widehat{\alpha}_{\text{UN}}^s$ satisfies:

$$\frac{1 - g^{1s}(\theta_{\text{UN}}^s(\widehat{\alpha}_{\text{UN}}^s))}{g^{0s}(\theta_{\text{UN}}^s(\widehat{\alpha}_{\text{UN}}^s))} = \frac{1 - (T_{11}^s(1 - \mathbb{G}_1^s(\theta_{\text{UN}}^s(\widehat{\alpha}_{\text{UN}}^s))) + T_{10}^s\mathbb{G}_1^s(\theta_{\text{UN}}^s(\widehat{\alpha}_{\text{UN}}^s)))}{T_{01}^s(1 - \mathbb{G}_0^s(\theta_{\text{UN}}^s(\widehat{\alpha}_{\text{UN}}^s))) + T_{00}^s\mathbb{G}_0^s(\theta_{\text{UN}}^s(\widehat{\alpha}_{\text{UN}}^s))} = \frac{1}{\widehat{\alpha}_{\text{UN}}^s} - 1.$$

One solution to the above equation is:

$$\widehat{\alpha}_{\text{UN}}^s = T_{11}^s(1 - \mathbb{G}_1^s(\theta_{\text{UN}}^s(\widehat{\alpha}_{\text{UN}}^s))) + T_{10}^s\mathbb{G}_1^s(\theta_{\text{UN}}^s(\widehat{\alpha}_{\text{UN}}^s)) = T_{01}^s(1 - \mathbb{G}_0^s(\theta_{\text{UN}}^s(\widehat{\alpha}_{\text{UN}}^s))) + T_{00}^s\mathbb{G}_0^s(\theta_{\text{UN}}^s(\widehat{\alpha}_{\text{UN}}^s))$$

It shows that $\widehat{\alpha}_{\text{UN}}^s$ is a convex combination of $T_{00}^s$, $T_{01}^s$, and also a convex combination of $T_{10}^s$, $T_{11}^s$.

$\forall \alpha_{\text{UN}}$ and $\mathbb{G}_0^s(x)$, $\mathbb{G}_1^s(x)$, there is a set of transitions with $T_{00}^s < \alpha_{\text{UN}} < T_{01}^s$ and $T_{10}^s < \alpha_{\text{UN}} < T_{11}^s$ (satisfy Condition 1(B)), or $T_{01}^s < \alpha_{\text{UN}} < T_{00}^s$ and $T_{11}^s < \alpha_{\text{UN}} < T_{10}^s$ (satisfy Condition 1(A)), such that the above equation holds with $\widehat{\alpha}_{\text{UN}}^s = \alpha_{\text{UN}}$, $\forall s \in \{a, b\}$, i.e., equitable equilibrium is attained.

Next we show that if $G_y^a(x) \neq G_y^b(x)$, then $\widehat{\alpha}_\mathcal{C}^b \neq \widehat{\alpha}_\mathcal{C}^a$ under these sets of transitions. Under the conditions of Theorem 2, $(\widehat{\alpha}_\mathcal{C}^a, \widehat{\alpha}_\mathcal{C}^b)$ is the intersection of two curves $\mathcal{C}_1 = \{(\alpha^a, \alpha^b) : \alpha^a = \psi_\mathcal{C}^a(\alpha^b), \alpha^b \in [0, 1]\}$ and $\mathcal{C}_2 = \{(\alpha^a, \alpha^b) : \alpha^b = \psi_\mathcal{C}^b(\alpha^a), \alpha^a \in [0, 1]\}$; furthermore, let $\widetilde{\alpha}_\mathcal{C}^a, \widetilde{\alpha}_\mathcal{C}^b$ be defined such that $\widetilde{\alpha}_\mathcal{C}^a = \psi_\mathcal{C}^a(\widetilde{\alpha}_\mathcal{C}^a), \widetilde{\alpha}_\mathcal{C}^b = \psi_\mathcal{C}^b(\widetilde{\alpha}_\mathcal{C}^b)$, which are the intersections of $\alpha^a = \psi_\mathcal{C}^a(\alpha^b)$ and $\alpha^a = \alpha^b$, as well as $\alpha^b = \psi_\mathcal{C}^b(\alpha^a)$ and $\alpha^a = \alpha^b$, respectively. Then in order to prove $\widehat{\alpha}_\mathcal{C}^b \neq \widehat{\alpha}_\mathcal{C}^a$, it is sufficient to show $\widetilde{\alpha}_\mathcal{C}^a \neq \widetilde{\alpha}_\mathcal{C}^b$.

Given $\alpha^a = \alpha^b = \alpha_{\text{UN}}$, because $G_y^a(x) \neq G_y^b(x)$, we have $\theta_{\text{UN}}^s(\alpha_{\text{UN}}) \neq \theta_\mathcal{C}^s(\alpha_{\text{UN}}, \alpha_{\text{UN}})$ and to satisfy Eqn. (3), there are only two possibilities: (1) $\theta_{\text{UN}}^a(\alpha_{\text{UN}}) > \theta_\mathcal{C}^a(\alpha_{\text{UN}}, \alpha_{\text{UN}})$, $\theta_{\text{UN}}^b(\alpha_{\text{UN}}) < \theta_\mathcal{C}^b(\alpha_{\text{UN}}, \alpha_{\text{UN}})$; (2) $\theta_{\text{UN}}^a(\alpha_{\text{UN}}) < \theta_\mathcal{C}^a(\alpha_{\text{UN}}, \alpha_{\text{UN}})$, $\theta_{\text{UN}}^b(\alpha_{\text{UN}}) > \theta_\mathcal{C}^b(\alpha_{\text{UN}}, \alpha_{\text{UN}})$.

WLOG, suppose the first case holds. Under Condition 1(B),

$$\frac{1 - g^{1b}(\theta_{\text{UN}}^b(\alpha_{\text{UN}}))}{g^{0b}(\theta_{\text{UN}}^b(\alpha_{\text{UN}}))} < \frac{1 - g^{1b}(\theta_\mathcal{C}^b(\alpha_{\text{UN}}, \alpha_{\text{UN}}))}{g^{0b}(\theta_\mathcal{C}^b(\alpha_{\text{UN}}, \alpha_{\text{UN}}))}; \quad \frac{1 - g^{1a}(\theta_{\text{UN}}^a(\alpha_{\text{UN}}))}{g^{0a}(\theta_{\text{UN}}^a(\alpha_{\text{UN}}))} > \frac{1 - g^{1a}(\theta_\mathcal{C}^a(\alpha_{\text{UN}}, \alpha_{\text{UN}}))}{g^{0a}(\theta_\mathcal{C}^a(\alpha_{\text{UN}}, \alpha_{\text{UN}}))}$$

It implies that $\widetilde{\alpha}_\mathcal{C}^b < \widehat{\alpha}_{\text{UN}}^b = \widehat{\alpha}_{\text{UN}}^a < \widetilde{\alpha}_\mathcal{C}^a$. Similarly, under Condition 1(A), $\widetilde{\alpha}_\mathcal{C}^b > \widehat{\alpha}_{\text{UN}}^b = \widehat{\alpha}_{\text{UN}}^a > \widetilde{\alpha}_\mathcal{C}^a$. Therefore, $\widehat{\alpha}_\mathcal{C}^a \neq \widehat{\alpha}_\mathcal{C}^b$.

In contrast, if $G_y^a(x) = G_y^b(x)$, we have $\theta_{\text{UN}}^s(\alpha) = \theta_\mathcal{C}^s(\alpha, \alpha)$ and $\widetilde{\alpha}_\mathcal{C}^b = \widetilde{\alpha}_\mathcal{C}^a$. Therefore, $\widehat{\alpha}_\mathcal{C}^a = \widehat{\alpha}_\mathcal{C}^b$. $\square$

**The proof of Theorem 4.**

*Proof.* WLOG, suppose that $\widehat{\alpha}_{\text{UN}}^a > \widehat{\alpha}_{\text{UN}}^b$ in the proof. Let $\psi_{\mathcal{C}}^a(\cdot), \psi_{\mathcal{C}}^b(\cdot)$ be balanced functions as defined in Theorem 2 under constraint $\mathcal{C}$. Firstly, we show that $\widehat{\alpha}_{\text{UN}}^b$ and $\widehat{\alpha}_{\text{UN}}^a$ are solutions to
$$\begin{cases} \alpha^b = \psi_{\mathcal{C}}^b(\alpha^a) \\ \alpha^a = \alpha^b \end{cases} \quad \text{and} \quad \begin{cases} \alpha^a = \psi_{\mathcal{C}}^a(\alpha^b) \\ \alpha^a = \alpha^b \end{cases}, \text{ respectively, i.e., } \widehat{\alpha}_{\text{UN}}^b = \psi_{\mathcal{C}}^b(\widehat{\alpha}_{\text{UN}}^b) \text{ and } \widehat{\alpha}_{\text{UN}}^a = \psi_{\mathcal{C}}^a(\widehat{\alpha}_{\text{UN}}^a).$$

Because $G_y^a(x) = G_y^b(x)$, $\forall y \in \{0,1\}, \forall x$, when $\alpha^a = \alpha^b = \alpha$, we have $\gamma^a(x) = \gamma^b(x)$, $\mathcal{P}_{\text{EqOpt}}^a(x) = \mathcal{P}_{\text{EqOpt}}^b(x)$ and $\mathcal{P}_{\text{DP}}^a(x) = \mathcal{P}_{\text{DP}}^b(x)$, which implies $\theta_{\mathcal{C}}^a(\alpha, \alpha) = \theta_{\mathcal{C}}^b(\alpha, \alpha)$; furthermore, the optimal fair policies of DP and EqOpt satisfy $\gamma^a(\theta_{\mathcal{C}}^a(\alpha, \alpha)) = \gamma^b(\theta_{\mathcal{C}}^b(\alpha, \alpha)) = \frac{u_-}{u_+ + u_-}$ according to the optimal fair policy equation:

$$\frac{p_a\alpha^a}{\gamma^a(\theta_{\text{EqOpt}}^a)} + \frac{p_b\alpha^b}{\gamma^b(\theta_{\text{EqOpt}}^b)} = \frac{p_a\alpha^a}{\frac{u_-}{u_+ + u_-}} + \frac{p_b\alpha^b}{\frac{u_-}{u_+ + u_-}}; \quad p_a\gamma^a(\theta_{\text{DP}}^a) + p_b\gamma^b(\theta_{\text{DP}}^b) = \frac{u_-}{u_+ + u_-}.$$

Because $\gamma^a(\theta_{\text{UN}}^a(\alpha)) = \gamma^b(\theta_{\text{UN}}^b(\alpha)) = \frac{u_-}{u_+ + u_-}$ we have $\gamma^a(\theta_{\text{UN}}^a(\alpha)) = \gamma^a(\theta_{\mathcal{C}}^a(\alpha, \alpha)) = \gamma^b(\theta_{\text{UN}}^b(\alpha)) = \gamma^b(\theta_{\mathcal{C}}^b(\alpha, \alpha))$ so that $\theta_{\mathcal{C}}^a(\alpha, \alpha) = \theta_{\text{UN}}^a(\alpha) = \theta_{\mathcal{C}}^b(\alpha, \alpha) = \theta_{\text{UN}}^b(\alpha)$ holds under any $\alpha$. $\forall s \in \{a, b\}$, because $\widehat{\alpha}_{\text{UN}}^s$ is the solution to balanced equation, i.e., $l^s(\widehat{\alpha}_{\text{UN}}^s) = h^s(\theta_{\text{UN}}^s(\widehat{\alpha}_{\text{UN}}^s))$. We have $l^s(\widehat{\alpha}_{\text{UN}}^s) = h^s(\theta_{\mathcal{C}}^s(\widehat{\alpha}_{\text{UN}}^s, \widehat{\alpha}_{\text{UN}}^s))$, which further implies $\widehat{\alpha}_{\text{UN}}^s = \psi_{\mathcal{C}}^s(\widehat{\alpha}_{\text{UN}}^s)$.

Under Condition 1(B), according to the proof of Theorem 2, we know that $0 \leq \frac{d\psi_{\mathcal{C}}^b(\alpha^a)}{d\alpha^a} < 1$ and $0 \leq \frac{d\psi_{\mathcal{C}}^a(\alpha^b)}{d\alpha^b} < 1$. Because $\widehat{\alpha}_{\text{UN}}^b = \psi_{\mathcal{C}}^b(\widehat{\alpha}_{\text{UN}}^b) < \widehat{\alpha}_{\text{UN}}^a = \psi_{\mathcal{C}}^a(\widehat{\alpha}_{\text{UN}}^a)$, we have $\widehat{\alpha}_{\text{UN}}^b < \psi_{\mathcal{C}}^b(\alpha^a) < \alpha^a$, $\forall \alpha^a \in [\widehat{\alpha}_{\text{UN}}^b, \widehat{\alpha}_{\text{UN}}^a]$. Similarly, we have $\alpha^b < \psi_{\mathcal{C}}^a(\alpha^b) < \widehat{\alpha}_{\text{UN}}^a$, $\forall \alpha^b \in [\widehat{\alpha}_{\text{UN}}^b, \widehat{\alpha}_{\text{UN}}^a]$. Therefore, after representing the two balanced functions as two curves $\mathcal{C}_1 = \{(\alpha^a, \alpha^b) : \alpha^a = \psi_{\mathcal{C}}^a(\alpha^b), \alpha^b \in [0, 1]\}$ and $\mathcal{C}_2 = \{(\alpha^a, \alpha^b) : \alpha^b = \psi_{\mathcal{C}}^b(\alpha^a), \alpha^a \in [0, 1]\}$ on the 2D plane $\{(\alpha^a, \alpha^b) : \alpha^a \in [0, 1], \alpha^b \in [0, 1]\}$, the intersection $(\widehat{\alpha}_{\mathcal{C}}^a, \widehat{\alpha}_{\mathcal{C}}^b)$ of $\mathcal{C}_1$ and $\mathcal{C}_2$ satisfies: 1) $\widehat{\alpha}_{\mathcal{C}}^a > \widehat{\alpha}_{\mathcal{C}}^b$; 2) $\widehat{\alpha}_{\text{UN}}^b < \widehat{\alpha}_{\mathcal{C}}^a < \widehat{\alpha}_{\text{UN}}^a$; 3) $\widehat{\alpha}_{\text{UN}}^b < \widehat{\alpha}_{\mathcal{C}}^b < \widehat{\alpha}_{\text{UN}}^a$. Therefore, $|\widehat{\alpha}_{\mathcal{C}}^a - \widehat{\alpha}_{\mathcal{C}}^b| \leq |\widehat{\alpha}_{\text{UN}}^a - \widehat{\alpha}_{\text{UN}}^b|$.

Under Condition 1(A), according to the proof of Theorem 2, we know that $-1 < \frac{d\psi_{\mathcal{C}}^b(\alpha^a)}{d\alpha^a} \leq 0$ and $-1 < \frac{d\psi_{\mathcal{C}}^a(\alpha^b)}{d\alpha^b} \leq 0$. Because $\widehat{\alpha}_{\text{UN}}^b = \psi_{\mathcal{C}}^b(\widehat{\alpha}_{\text{UN}}^b) < \widehat{\alpha}_{\text{UN}}^a = \psi_{\mathcal{C}}^a(\widehat{\alpha}_{\text{UN}}^a)$, we have $\psi_{\mathcal{C}}^b(\alpha^a) < \widehat{\alpha}_{\text{UN}}^b$, $\forall \alpha^a > \widehat{\alpha}_{\text{UN}}^b$. Similarly, we have $\psi_{\mathcal{C}}^a(\alpha^b) > \widehat{\alpha}_{\text{UN}}^a$, $\forall \alpha^b < \widehat{\alpha}_{\text{UN}}^a$. Due to the existence of equilibrium, the intersection $(\widehat{\alpha}_{\mathcal{C}}^a, \widehat{\alpha}_{\mathcal{C}}^b)$ of $\mathcal{C}_1$ and $\mathcal{C}_2$ must satisfy: 1) $\widehat{\alpha}_{\mathcal{C}}^a > \widehat{\alpha}_{\mathcal{C}}^b$; 2) $\widehat{\alpha}_{\text{UN}}^a < \widehat{\alpha}_{\mathcal{C}}^a$; 3) $\widehat{\alpha}_{\mathcal{C}}^b < \widehat{\alpha}_{\text{UN}}^b$. Therefore, $|\widehat{\alpha}_{\mathcal{C}}^a - \widehat{\alpha}_{\mathcal{C}}^b| \geq |\widehat{\alpha}_{\text{UN}}^a - \widehat{\alpha}_{\text{UN}}^b|$. $\qquad\square$

**The proof of Theorem 5.**

*Proof.* The proof is under the conditions of Theorem 2 such that there is unique equilibrium of qualification rate. Under fairness constraint $\mathcal{C} = \text{EqOpt}$ or DP, consider 2D plane $\{(\alpha^a, \alpha^b) : \alpha^a \in [0, 1], \alpha^b \in [0, 1]\}$, and note that equilibrium $(\widehat{\alpha}_{\mathcal{C}}^a, \widehat{\alpha}_{\mathcal{C}}^b)$ is the intersection of two curves $\mathcal{C}_1 = \{(\alpha^a, \alpha^b) : \alpha^a = \psi_{\mathcal{C}}^a(\alpha^b), \alpha^b \in [0, 1]\}$ and $\mathcal{C}_2 = \{(\alpha^a, \alpha^b) : \alpha^b = \psi_{\mathcal{C}}^b(\alpha^a), \alpha^a \in [0, 1]\}$. Consider a line $\{(\alpha^a, \alpha^b) : \alpha^a = \alpha^b, \alpha^a \in [0, 1], \alpha^b \in [0, 1]\}$, which has unique intersection $\widetilde{\alpha}_{\mathcal{C}}^a$ with $\mathcal{C}_1$, and unique intersection $\widetilde{\alpha}_{\mathcal{C}}^b$ with $\mathcal{C}_2$. That is, $\widetilde{\alpha}_{\mathcal{C}}^a = \psi_{\mathcal{C}}^a(\widetilde{\alpha}_{\mathcal{C}}^a)$, $\widetilde{\alpha}_{\mathcal{C}}^b = \psi_{\mathcal{C}}^b(\widetilde{\alpha}_{\mathcal{C}}^b)$.

First of all, we show that *if $\frac{u_+}{u_-} \geq \frac{1 - T_{10}}{T_{00}}\beta(\widehat{x})$, under Condition 1(B), $\widehat{\alpha}_{UN}^b < \widehat{\alpha}_{UN}^a$.*

By Condition 2, given any $\alpha^a = \alpha^b = \alpha$, the corresponding qualification profiles of $\mathcal{G}_a, \mathcal{G}_b$ satisfy the followings: $\gamma^b(\widehat{x}) = \gamma^a(\widehat{x})$; $\gamma^b(x) < \gamma^a(x), \forall x < \widehat{x}$; $\gamma^b(x) > \gamma^a(x), \forall x > \widehat{x}$. Let $\overline{\alpha}$ be qualification rate such that $\gamma^a(\widehat{x}) = \gamma^b(\widehat{x}) = \frac{u_-}{u_+ + u_-} \Longrightarrow \frac{u_+}{u_-} = \beta(\widehat{x})(\frac{1}{\overline{\alpha}} - 1)$, where $\beta(\widehat{x}) := \frac{G_0^a(\widehat{x})}{G_1^a(\widehat{x})} = \frac{G_0^b(\widehat{x})}{G_1^b(\widehat{x})}$, then $\forall \alpha \in [\overline{\alpha}, 1]$, $\gamma^a(\theta_{\text{UN}}^a(\alpha)) = \gamma^b(\theta_{\text{UN}}^b(\alpha)) = \frac{u_-}{u_+ + u_-} < \frac{1}{\beta(\widehat{x})(\frac{1}{\overline{\alpha}} - 1) + 1} = \gamma^a(\widehat{x}) = \gamma^b(\widehat{x})$. Thus, $\forall \alpha \in [\overline{\alpha}, 1]$, $\theta_{\text{UN}}^a(\alpha) < \theta_{\text{UN}}^b(\alpha) < \widehat{x}$, which implies $\mathbb{G}_1^a(\theta_{\text{UN}}^a(\alpha)) < \mathbb{G}_1^b(\theta_{\text{UN}}^b(\alpha))$ and $\mathbb{G}_0^a(\theta_{\text{UN}}^a(\alpha)) < \mathbb{G}_0^b(\theta_{\text{UN}}^b(\alpha))$; furthermore, under Condition 1(B), we have

$$\frac{1 - T_{11}}{T_{01}} < \frac{1 - g^{1a}(\theta_{\text{UN}}^a(\alpha))}{g^{0a}(\theta_{\text{UN}}^a(\alpha))} < \frac{1 - g^{1b}(\theta_{\text{UN}}^b(\alpha))}{g^{0b}(\theta_{\text{UN}}^b(\alpha))} < \frac{1 - T_{10}}{T_{00}}, \; \forall \alpha \in [\overline{\alpha}, 1].$$

Because $\widehat{\alpha}_{\text{UN}}^a$ and $\widehat{\alpha}_{\text{UN}}^b$ are solutions to balance equations, i.e., $\frac{1}{\widehat{\alpha}_{\text{UN}}^a} - 1 = \frac{1-g^{1a}(\theta_{\text{UN}}^a(\widehat{\alpha}_{\text{UN}}^a))}{g^{0a}(\theta_{\text{UN}}^a(\widehat{\alpha}_{\text{UN}}^a))}$, $\frac{1}{\widehat{\alpha}_{\text{UN}}^b} - 1 = \frac{1-g^{1b}(\theta_{\text{UN}}^b(\widehat{\alpha}_{\text{UN}}^b))}{g^{0b}(\theta_{\text{UN}}^b(\widehat{\alpha}_{\text{UN}}^b))}$. If $\overline{\alpha} \le \widehat{\alpha}_{\text{UN}}^b$, the $\widehat{\alpha}_{\text{UN}}^b < \widehat{\alpha}_{\text{UN}}^a$ must hold under Condition 1(B). Next, we show that a sufficient condition of $\overline{\alpha} \le \widehat{\alpha}_{\text{UN}}^b$ is $\frac{u_+}{u_-} \ge \frac{1-T_{10}}{T_{00}}\beta(\widehat{x})$.

$\frac{u_+}{u_-} \ge \frac{1-T_{10}}{T_{00}}\beta(\widehat{x}) \implies \frac{1}{\overline{\alpha}} - 1 \ge \frac{1-T_{10}}{T_{00}}$. Since $\frac{1}{\widehat{\alpha}_{\text{UN}}^b} - 1 < \frac{1-T_{10}}{T_{00}}$, we have $\frac{1}{\widehat{\alpha}_{\text{UN}}^b} - 1 < \frac{1}{\overline{\alpha}} - 1$. Thus, $\overline{\alpha} \le \widehat{\alpha}_{\text{UN}}^b$. Therefore, if $\frac{u_+}{u_-} \ge \frac{1-T_{10}}{T_{00}}\beta(\widehat{x})$, under Condition 1(B), $\widehat{\alpha}_{\text{UN}}^b < \widehat{\alpha}_{\text{UN}}^a$.

**Fairness constraint** EqOpt. Secondly, we show that for EqOpt fair policy, if $\frac{u_+}{u_-} \ge \frac{1-T_{10}}{T_{00}}\beta(\widehat{x})$, under Condition 1(B), $\widehat{\alpha}_{\text{UN}}^a - \widehat{\alpha}_{\text{UN}}^b > \widehat{\alpha}_{\text{EqOpt}}^a - \widehat{\alpha}_{\text{EqOpt}}^b \ge 0$. Because two curves $\mathcal{C}_1, \mathcal{C}_2$ are monotonic increasing. It's sufficient to show two parts: (1) $\widetilde{\alpha}_{\text{EqOpt}}^a < \widehat{\alpha}_{\text{UN}}^a$, $\widetilde{\alpha}_{\text{EqOpt}}^b > \widehat{\alpha}_{\text{UN}}^b$; (2) $\widetilde{\alpha}_{\text{EqOpt}}^a \ge \widetilde{\alpha}_{\text{EqOpt}}^b$.

Under EqOpt constraint, $\forall \alpha^a, \alpha^b$, $\mathbb{G}_1^a(\theta_{\text{EqOpt}}^a(\alpha^a, \alpha^b)) = \mathbb{G}_1^b(\theta_{\text{EqOpt}}^b(\alpha^a, \alpha^b))$ must hold so that $\theta_{\text{EqOpt}}^a(\alpha^a, \alpha^b) = \theta_{\text{EqOpt}}^b(\alpha^a, \alpha^b)$. Consider the case $\alpha^a = \alpha^b = \alpha$, $\forall \alpha \ge \overline{\alpha}$, we have $\theta_{\text{EqOpt}}^a(\alpha, \alpha) = \theta_{\text{EqOpt}}^b(\alpha, \alpha)$ and $\theta_{\text{UN}}^a(\alpha) < \theta_{\text{UN}}^b(\alpha)$. It implies that $\theta_{\text{UN}}^a(\alpha) < \theta_{\text{EqOpt}}^a(\alpha, \alpha) = \theta_{\text{EqOpt}}^b(\alpha, \alpha) < \theta_{\text{UN}}^b(\alpha) < \widehat{x}$, otherwise Equation (3) will be violated. Therefore, the followings hold $\forall \alpha \in [\overline{\alpha}, 1]$,

$$\frac{1 - g^{1a}(\theta_{\text{EqOpt}}^a(\alpha, \alpha))}{g^{0a}(\theta_{\text{EqOpt}}^a(\alpha, \alpha))} > \frac{1 - g^{1a}(\theta_{\text{UN}}^a(\alpha))}{g^{0a}(\theta_{\text{UN}}^a(\alpha))}; \quad \frac{1 - g^{1b}(\theta_{\text{EqOpt}}^b(\alpha, \alpha))}{g^{0b}(\theta_{\text{EqOpt}}^b(\alpha, \alpha))} < \frac{1 - g^{1b}(\theta_{\text{UN}}^b(\alpha))}{g^{0b}(\theta_{\text{UN}}^b(\alpha))}.$$

$\forall s \in \{a, b\}$, $\widetilde{\alpha}_{\text{EqOpt}}^s$ is the solution to $\frac{1-g^{1s}(\theta_{\text{EqOpt}}^s(\alpha, \alpha))}{g^{0s}(\theta_{\text{EqOpt}}^s(\alpha, \alpha))} = \frac{1}{\alpha} - 1$ while $\widehat{\alpha}_{\text{UN}}^s$ is the solution to $\frac{1-g^{1s}(\theta_{\text{UN}}^s(\alpha))}{g^{0s}(\theta_{\text{UN}}^s(\alpha))} = \frac{1}{\alpha} - 1$. Since $\overline{\alpha} \le \widehat{\alpha}_{\text{UN}}^b < \widehat{\alpha}_{\text{UN}}^a$, it implies $\widetilde{\alpha}_{\text{EqOpt}}^a < \widehat{\alpha}_{\text{UN}}^a$, $\widetilde{\alpha}_{\text{EqOpt}}^b > \widehat{\alpha}_{\text{UN}}^b$.

Next, show that $\widetilde{\alpha}_{\text{EqOpt}}^a \ge \widetilde{\alpha}_{\text{EqOpt}}^b$. $\forall \alpha \ge \overline{\alpha}$, $\theta_{\text{EqOpt}}^a(\alpha, \alpha) = \theta_{\text{EqOpt}}^b(\alpha, \alpha)$ implies $\mathbb{G}_1^a(\theta_{\text{EqOpt}}^a(\alpha, \alpha)) = \mathbb{G}_1^b(\theta_{\text{EqOpt}}^b(\alpha, \alpha))$ and $\mathbb{G}_0^a(\theta_{\text{EqOpt}}^a(\alpha, \alpha)) \le \mathbb{G}_0^b(\theta_{\text{EqOpt}}^b(\alpha, \alpha))$. Therefore,

$$\frac{1 - g^{1a}(\theta_{\text{EqOpt}}^a(\alpha, \alpha))}{g^{0a}(\theta_{\text{EqOpt}}^a(\alpha, \alpha))} \le \frac{1 - g^{1b}(\theta_{\text{EqOpt}}^b(\alpha, \alpha))}{g^{0b}(\theta_{\text{EqOpt}}^b(\alpha, \alpha))}.$$

Intersections with function $\frac{1}{\alpha} - 1$ satisfies $\widetilde{\alpha}_{\text{EqOpt}}^a \ge \widetilde{\alpha}_{\text{EqOpt}}^b$.

It thus concludes that $\widehat{\alpha}_{\text{UN}}^a - \widehat{\alpha}_{\text{UN}}^b > \widehat{\alpha}_{\text{EqOpt}}^a - \widehat{\alpha}_{\text{EqOpt}}^b \ge 0$.

**Fairness constraint** DP. Finally, consider DP fair policy, where $\forall \alpha^a, \alpha^b$, $(1-\alpha^a)\mathbb{G}_0^a(\theta_{\text{DP}}^a(\alpha^a, \alpha^b)) + \alpha^a\mathbb{G}_1^a(\theta_{\text{DP}}^a(\alpha^a, \alpha^b)) = (1 - \alpha^b)\mathbb{G}_0^b(\theta_{\text{DP}}^b(\alpha^a, \alpha^b)) + \alpha^b\mathbb{G}_1^b(\theta_{\text{DP}}^b(\alpha^a, \alpha^b))$ must hold.

We first show that under Condition 1(B), $\widetilde{\alpha}_{\text{DP}}^a < \widehat{\alpha}_{\text{UN}}^a$, $\widetilde{\alpha}_{\text{DP}}^b > \widehat{\alpha}_{\text{UN}}^b$. Consider the case $\alpha^a = \alpha^b = \alpha$, $\forall \alpha \ge \overline{\alpha}$. Since $\forall x$, $(1 - \alpha)\mathbb{G}_0^b(x) + \alpha\mathbb{G}_1^b(x) \ge (1 - \alpha)\mathbb{G}_0^a(x) + \alpha\mathbb{G}_1^a(x)$, $(1 - \alpha)\mathbb{G}_0^a(\theta_{\text{DP}}^a(\alpha, \alpha)) + \alpha\mathbb{G}_1^a(\theta_{\text{DP}}^a(\alpha, \alpha)) = (1-\alpha)\mathbb{G}_0^b(\theta_{\text{DP}}^b(\alpha, \alpha)) + \alpha\mathbb{G}_1^b(\theta_{\text{DP}}^b(\alpha, \alpha))$ implies $\theta_{\text{DP}}^a(\alpha, \alpha) \ge \theta_{\text{DP}}^b(\alpha, \alpha)$. Because $\theta_{\text{UN}}^a(\alpha) < \theta_{\text{UN}}^b(\alpha)$, $\forall \alpha \ge \overline{\alpha}$. It implies that $\theta_{\text{DP}}^a(\alpha, \alpha) > \theta_{\text{UN}}^a(\alpha)$ and $\widehat{x} > \theta_{\text{UN}}^b(\alpha) > \theta_{\text{DP}}^b(\alpha, \alpha)$ must hold. Therefore, $\forall \alpha \in [\overline{\alpha}, 1]$,

$$\frac{1 - g^{1a}(\theta_{\text{DP}}^a(\alpha, \alpha))}{g^{0a}(\theta_{\text{DP}}^a(\alpha, \alpha))} > \frac{1 - g^{1a}(\theta_{\text{UN}}^a(\alpha))}{g^{0a}(\theta_{\text{UN}}^a(\alpha))}; \quad \frac{1 - g^{1b}(\theta_{\text{DP}}^b(\alpha, \alpha))}{g^{0b}(\theta_{\text{DP}}^b(\alpha, \alpha))} < \frac{1 - g^{1b}(\theta_{\text{UN}}^b(\alpha))}{g^{0b}(\theta_{\text{UN}}^b(\alpha))}$$

Similar to reasoning in EqOpt case, we have $\widetilde{\alpha}_{\text{DP}}^a < \widehat{\alpha}_{\text{UN}}^a$, $\widetilde{\alpha}_{\text{DP}}^b > \widehat{\alpha}_{\text{UN}}^b$.

Different from EqOpt fairness where $\widetilde{\alpha}_{\text{EqOpt}}^a \ge \widetilde{\alpha}_{\text{EqOpt}}^b$, both $\widetilde{\alpha}_{\text{DP}}^a \ge \widetilde{\alpha}_{\text{DP}}^b$ and $\widetilde{\alpha}_{\text{DP}}^a \le \widetilde{\alpha}_{\text{DP}}^b$ are likely to occur, depending on distributions $G_0^a(x)$, $G_0^b(x)$, $G_1^a(x)$ and $G_1^b(x)$. It is because $\theta_{\text{DP}}^a(\alpha, \alpha) > \theta_{\text{DP}}^b(\alpha, \alpha)$ can result in either $\mathbb{G}_0^a(\theta_{\text{DP}}^a(\alpha, \alpha)) \le \mathbb{G}_0^b(\theta_{\text{DP}}^b(\alpha, \alpha))$ or $\mathbb{G}_0^a(\theta_{\text{DP}}^a(\alpha, \alpha)) \ge \mathbb{G}_0^b(\theta_{\text{DP}}^b(\alpha, \alpha))$.

For these two outcomes, if $\widetilde{\alpha}_{\text{DP}}^a \ge \widetilde{\alpha}_{\text{DP}}^b$, then DP fair policy results in a more equitable equilibrium than unconstrained policy; if $\widetilde{\alpha}_{\text{DP}}^a \le \widetilde{\alpha}_{\text{DP}}^b$, it means the disadvantaged group is flipped from $\mathcal{G}_b$ to $\mathcal{G}_a$.

$\square$

**The proof of Proposition 1.**

*Proof.* In the proof, we simplify the notations by removing subscript $\mathcal{C}$.

Let $\psi^s(\cdot)$, $\psi^{s'}(\cdot)$ be balanced function of policies $(\theta^a, \theta^b)$ and $(\theta^{a'}, \theta^{b'})$, respectively.

According to the balanced equation (5),

$$\frac{1}{\alpha^s} - 1 = \frac{1 - g^{1s}(\theta^s(\alpha^a, \alpha^b))}{g^{0s}(\theta^s(\alpha^a, \alpha^b))} = \frac{1 - (T_{11}^s(1 - \mathbb{G}_1^s(\theta^s(\alpha^a, \alpha^b))) + T_{10}^s\mathbb{G}_1^s(\theta^s(\alpha^a, \alpha^b)))}{T_{01}^s(1 - \mathbb{G}_0^s(\theta^s(\alpha^a, \alpha^b))) + T_{00}^s\mathbb{G}_0^s(\theta^s(\alpha^a, \alpha^b))}.$$

Under Condition (B), $\forall \alpha^a, \alpha^b \in [0,1]$, $\theta^{a'}(\alpha^a, \alpha^b) < \theta^a(\alpha^a, \alpha^b)$ and $\theta^{b'}(\alpha^a, \alpha^b) < \theta^b(\alpha^a, \alpha^b)$.

Under Condition (A), $\forall \alpha^a, \alpha^b \in [0,1]$, $\theta^{a'}(\alpha^a, \alpha^b) > \theta^a(\alpha^a, \alpha^b)$ and $\theta^{b'}(\alpha^a, \alpha^b) > \theta^b(\alpha^a, \alpha^b)$.

Both imply that $\frac{1 - g^{1s}(\theta^s(\alpha^a, \alpha^b))}{g^{0s}(\theta^s(\alpha^a, \alpha^b))} > \frac{1 - g^{1s}(\theta^{s'}(\alpha^a, \alpha^b))}{g^{0s}(\theta^{s'}(\alpha^a, \alpha^b))}$, and $\forall \alpha^a, \alpha^b \in [0,1]$, $\psi^a(\alpha^b) < \psi^{a'}(\alpha^b)$ and $\psi^b(\alpha^a) < \psi^{b'}(\alpha^a)$ hold. As a consequence, $\widehat{\alpha}^{a'} > \widehat{\alpha}^a$ and $\widehat{\alpha}^{b'} > \widehat{\alpha}^b$.

Now consider the long-run average utility of institute $\overline{U}(\theta^a, \theta^b) = \lim_{T\to\infty} \frac{1}{T}\sum_{t=1}^T \mathcal{U}_t(\theta^a, \theta^b)$, where the instantaneous utility at $t$ under threshold policies $\theta^a, \theta^b$ is

$$
\begin{aligned}
\mathcal{U}_t(\theta^a, \theta^b) &= \sum_{s=a,b} p_s \mathbb{E}_{X_t|S=s}[\mathbf{1}(X_t \geq \theta^s)(\gamma_t^s(X_t)(u_+ + u_-) - u_-)] \\
&= \sum_{s=a,b} p_s \int_{\theta^s}^\infty (\gamma_t^s(x)(u_+ + u_-) - u_-)\mathbb{P}(X_t = x \mid S = s)dx \\
&= \sum_{s=a,b} p_s \int_{\theta^s}^\infty \alpha_t^s\big(G_1^s(x)u_+ + G_0^s(x)u_-\big) - G_0^s(x)u_- dx
\end{aligned}
$$

In the followings, we use a special case ($\mathcal{C} = \texttt{EqOpt}$, $G_y^a(x) = G_y^b(x)$, $\forall x, y = 0, 1$, under Condition 1(B)) to show that $\overline{U}(\theta^{a'}, \theta^{b'}) > \overline{U}(\theta^a, \theta^b)$ can be attained, i.e., the long-run average utility under policy $(\theta^{a'}, \theta^{b'})$ can be higher than myopic optimal policy $(\theta^a, \theta^b)$.

Since the qualification rates of two groups converge to equilibrium, $\overline{U}(\theta^a, \theta^b) = \mathcal{U}_\infty(\theta^a, \theta^b)$ is the same as instantaneous expected utility of institute at the equilibrium state. To show that $\overline{U}(\theta^{a'}, \theta^{b'}) > \overline{U}(\theta^a, \theta^b)$, we prove the following holds,

$$\sum_{s=a,b} p_s \int_{\theta^{s'}(\widehat{\alpha}^{a'}, \widehat{\alpha}^{b'})}^\infty f(x; \widehat{\alpha}^{s'})dx > \sum_{s=a,b} p_s \int_{\theta^s(\widehat{\alpha}^a, \widehat{\alpha}^b)}^\infty f(x; \widehat{\alpha}^s)dx \tag{7}$$

where $f(x; \widehat{\alpha}^s) := \widehat{\alpha}^s\big(G_1^s(x)u_+ + G_0^s(x)u_-\big) - G_0^s(x)u_-$.

Because $\widehat{\alpha}^{s'} > \widehat{\alpha}^s$, $\theta^{s'}(\widehat{\alpha}^{a'}, \widehat{\alpha}^{b'}) < \theta^s(\widehat{\alpha}^{a'}, \widehat{\alpha}^{b'}) < \theta^s(\widehat{\alpha}^a, \widehat{\alpha}^b)$ holds under Condition (B). LHS of above inequality can be written as

$$\sum_{s=a,b} p_s\Big(\int_{\theta^{s'}(\widehat{\alpha}^{a'}, \widehat{\alpha}^{b'})}^{\theta^s(\widehat{\alpha}^a, \widehat{\alpha}^b)} f(x; \widehat{\alpha}^{s'})dx + \int_{\theta^s(\widehat{\alpha}^a, \widehat{\alpha}^b)}^\infty f(x; \widehat{\alpha}^{s'})dx\Big).$$

Inequality (7) can further be re-organized,

$$\sum_{s=a,b} p_s \int_{\theta^{s'}(\widehat{\alpha}^{a'}, \widehat{\alpha}^{b'})}^{\theta^s(\widehat{\alpha}^a, \widehat{\alpha}^b)} f(x; \widehat{\alpha}^{s'})dx > \sum_{s=a,b} p_s \int_{\theta^s(\widehat{\alpha}^a, \widehat{\alpha}^b)}^\infty \big(f(x; \widehat{\alpha}^s) - f(x; \widehat{\alpha}^{s'})\big)dx \tag{8}$$

Consider a special case where $\mathcal{C} = \texttt{EqOpt}$ and $G_y^a(x) = G_y^b(x) = G_y(x)$, $\forall x, \forall y \in \{0, 1\}$. Then $\forall \alpha^a, \alpha^b$, we have $\theta^a(\alpha^a, \alpha^b) = \theta^b(\alpha^a, \alpha^b)$ and $\theta^{a'}(\alpha^a, \alpha^b) = \theta^{b'}(\alpha^a, \alpha^b)$. Inequality (8) can be reduced to the following, $\forall s \in \{a, b\}$, simplify notations and let $\widehat{\theta} := \theta^s(\widehat{\alpha}^a, \widehat{\alpha}^b)$, $\widehat{\theta'} := \theta^{s'}(\widehat{\alpha}^{a'}, \widehat{\alpha}^{b'})$.

$$
\begin{aligned}
&\Big(p_a\widehat{\alpha}^{a'} + p_b\widehat{\alpha}^{b'}\Big)\Big(\mathbb{G}_1(\widehat{\theta}) - \mathbb{G}_1(\widehat{\theta'})\Big)u_+ \\
&+ \underbrace{\Big(u_+(1 - \mathbb{G}_1(\widehat{\theta})) + u_-(1 - \mathbb{G}_0(\widehat{\theta}))\Big)\Big(p_a(\widehat{\alpha}^{a'} - \widehat{\alpha}^a) + p_b(\widehat{\alpha}^{b'} - \widehat{\alpha}^b)\Big)}_{\textbf{term 1}} \\
&> \Big(p_a(1 - \widehat{\alpha}^{a'}) + p_b(1 - \widehat{\alpha}^{b'})\Big)\Big(\mathbb{G}_0(\widehat{\theta}) - \mathbb{G}_0(\widehat{\theta'})\Big)u_-
\end{aligned} \tag{9}
$$

Because $\frac{1}{\widehat{\alpha}^{s'}} - 1 = \frac{1-g^{1s}(\widehat{\theta}')}{g^{0s}(\widehat{\theta}')}$ and $\frac{1}{\widehat{\alpha}^{s}} - 1 = \frac{1-g^{1s}(\widehat{\theta})}{g^{0s}(\widehat{\theta})}$.

$$\widehat{\alpha}^{s'} - \widehat{\alpha}^{s} > \frac{T_{01}^{s} - T_{00}^{s}}{1 - T_{10}^{s} + T_{01}^{s}}(\mathbb{G}_0(\widehat{\theta}) - \mathbb{G}_0(\widehat{\theta}'))$$

We have **term 1** $>$

$$\underbrace{\left(\frac{u_+}{u_-}(1 - \mathbb{G}_1(\widehat{\theta})) + (1 - \mathbb{G}_0(\widehat{\theta}))\right)\left(p_a\frac{T_{01}^{a} - T_{00}^{a}}{1 - T_{10}^{a} + T_{01}^{a}} + p_b\frac{T_{01}^{b} - T_{00}^{b}}{1 - T_{10}^{b} + T_{01}^{b}}\right)\left(\mathbb{G}_0(\widehat{\theta}) - \mathbb{G}_0(\widehat{\theta}')\right)u_-}_{:=h(\widehat{\theta})>0}$$

For the optimal `EqOpt` fair threshold $\theta(\widehat{\alpha}^{a'}, \widehat{\alpha}^{b'})$, the following holds

$$\left(p_a\widehat{\alpha}^{a'} + p_b\widehat{\alpha}^{b'}\right)G_1(\theta(\widehat{\alpha}^{a'}, \widehat{\alpha}^{b'}))u_+ = \left(p_a(1 - \widehat{\alpha}^{a'}) + p_b(1 - \widehat{\alpha}^{b'})\right)G_0(\theta(\widehat{\alpha}^{a'}, \widehat{\alpha}^{b'}))u_-$$

$$\left(p_a\widehat{\alpha}^{a'} + p_b\widehat{\alpha}^{b'}\right)G_1(x)u_+ > \left(p_a(1 - \widehat{\alpha}^{a'}) + p_b(1 - \widehat{\alpha}^{b'})\right)G_0(x)u_-, \forall x > \theta(\widehat{\alpha}^{a'}, \widehat{\alpha}^{b'})$$

$$\left(p_a\widehat{\alpha}^{a'} + p_b\widehat{\alpha}^{b'}\right)G_1(x)u_+ < \left(p_a(1 - \widehat{\alpha}^{a'}) + p_b(1 - \widehat{\alpha}^{b'})\right)G_0(x)u_-, \forall x < \theta(\widehat{\alpha}^{a'}, \widehat{\alpha}^{b'})$$

It implies that $\exists$ some $\delta > 0$ s.t. $\forall x \in (\theta(\widehat{\alpha}^{a'}, \widehat{\alpha}^{b'}) - \delta, \theta(\widehat{\alpha}^{a'}, \widehat{\alpha}^{b'}) + \delta) := \mathcal{B}(\theta(\widehat{\alpha}^{a'}, \widehat{\alpha}^{b'}), \delta)$,

$$\left(p_a\widehat{\alpha}^{a'} + p_b\widehat{\alpha}^{b'}\right)G_1(x)u_+ + h(\widehat{\theta})G_0(x)u_- > \left(p_a(1 - \widehat{\alpha}^{a'}) + p_b(1 - \widehat{\alpha}^{b'})\right)G_0(x)u_-.$$

$\widehat{\theta}, \widehat{\theta}' \in \mathcal{B}(\theta(\widehat{\alpha}^{a'}, \widehat{\alpha}^{b'}), \delta)$ can be satisfied as long as $|\theta^s(\alpha^a, \alpha^b) - \theta^{s'}(\alpha^a, \alpha^b)| \leq \epsilon$ for some sufficiently small $\epsilon > 0$.

Using the mean value theorem, $\exists G_y(x)$ and $\widetilde{x} \in (\widehat{\theta}', \widehat{\theta}) \subset \mathcal{B}(\theta(\widehat{\alpha}^{a'}, \widehat{\alpha}^{b'}), \delta)$ s.t.

$$\begin{aligned}
& \left(p_a\widehat{\alpha}^{a'} + p_b\widehat{\alpha}^{b'}\right)(\mathbb{G}_1(\widehat{\theta}) - \mathbb{G}_1(\widehat{\theta}'))u_+ + h(\widehat{\theta})(\mathbb{G}_0(\widehat{\theta}) - \mathbb{G}_0(\widehat{\theta}'))u_- \\
= & \left(\left(p_a\widehat{\alpha}^{a'} + p_b\widehat{\alpha}^{b'}\right)G_1(\widetilde{x})u_+ + h(\widehat{\theta})G_0(\widetilde{x})u_-\right)(\widehat{\theta} - \widehat{\theta}') \\
> & \left(\left(p_a(1 - \widehat{\alpha}^{a'}) + p_b(1 - \widehat{\alpha}^{b'})\right)G_0(\widetilde{x})u_-\right)(\widehat{\theta} - \widehat{\theta}') \\
\geq & \left(p_a(1 - \widehat{\alpha}^{a'}) + p_b(1 - \widehat{\alpha}^{b'})\right)(\mathbb{G}_0(\widehat{\theta}) - \mathbb{G}_0(\widehat{\theta}'))u_-.
\end{aligned}$$

Therefore, inequality (9) holds and $\overline{U}(\theta^{a'}, \theta^{b'}) > \overline{U}(\theta^a, \theta^b)$. $\qquad\square$

**The proof of Proposition 2.**

*Proof.* To ensure $\alpha_t^s \to \widehat{\alpha}$, threshold policy $\theta^s(\alpha^s)$ as a function of $\alpha^s \in [0, 1]$ should be designed such that $\frac{1-g^{1s}(\theta^s(\alpha^s))}{g^{0s}(\theta^s(\alpha^s))} = \frac{1}{\alpha^s} - 1$ has a unique solution $\widehat{\alpha}$. Let $\mathcal{I}_s := \left[\frac{1-\max\{T_{11}^s, T_{10}^s\}}{\max\{T_{01}^s, T_{00}^s\}}, \frac{1-\min\{T_{11}^s, T_{10}^s\}}{\min\{T_{01}^s, T_{00}^s\}}\right]$, then $\frac{1-g^{1s}(\theta^s(\alpha^s))}{g^{0s}(\theta^s(\alpha^s))} \in \mathcal{I}_s$ for any threshold policy $\theta^s(\alpha^s)$.

If $\mathcal{I}_a \cap \mathcal{I}_b = \emptyset$, then $\frac{1-g^{1a}(\theta^a(\alpha))}{g^{0a}(\theta^a(\alpha))} = \frac{1-g^{1b}(\theta^b(\alpha))}{g^{0b}(\theta^b(\alpha))}$ can never be attained, i.e., no threshold policy can result in equitable equilibrium.

If $\mathcal{I}_a \cap \mathcal{I}_b \neq \emptyset$, then $\forall\widehat{\alpha} \in \mathcal{I}_a \cap \mathcal{I}_b$ and $\forall s \in \{a, b\}$, there exists threshold policy $\theta^s(\alpha^s)$ such that $\frac{1-g^{1s}(\theta^s(\widehat{\alpha}))}{g^{0s}(\theta^s(\widehat{\alpha}))} = \frac{1}{\widehat{\alpha}} - 1$. Specifically, under Condition 1(B) (resp. 1(A)), function

$$h^s(x) := \frac{1 - g^{1s}(x)}{g^{0s}(x)} = \frac{1 - (T_{11}^s(1 - \mathbb{G}_1^s(x)) + T_{10}^s\mathbb{G}_1^s(x))}{T_{01}^s(1 - \mathbb{G}_0^s(x)) + T_{00}^s\mathbb{G}_0^s(x)}$$

is strictly increasing (resp. decreasing) in $x \in (-\infty, +\infty)$ from $\frac{1-T_{11}^s}{T_{01}^s}$ (resp. $\frac{1-T_{10}^s}{T_{00}^s}$) to $\frac{1-T_{10}^s}{T_{00}^s}$ (resp. $\frac{1-T_{11}^s}{T_{01}^s}$) and any non-increasing function $\theta^s(\alpha^s)$ that satisfies $\theta^s(\widehat{\alpha}) = (h^s)^{-1}(\frac{1}{\widehat{\alpha}} - 1)$ can result in $\alpha_t^s \to \widehat{\alpha}$, where $(h^s)^{-1}(\cdot)$ is the inverse function of $h^s(\cdot)$. $\qquad\square$

**The proof of Proposition 3.**

*Proof.* According to the balanced equation (5),

$$\frac{1}{\alpha^s} - 1 = \frac{1 - g^{1s}(\theta^s(\alpha^a, \alpha^b))}{g^{0s}(\theta^s(\alpha^a, \alpha^b))} = \frac{1 - (T_{11}^s(1 - \mathbb{G}_1^s(\theta^s(\alpha^a, \alpha^b))) + T_{10}^s \mathbb{G}_1^s(\theta^s(\alpha^a, \alpha^b)))}{T_{01}^s(1 - \mathbb{G}_0^s(\theta^s(\alpha^a, \alpha^b))) + T_{00}^s \mathbb{G}_0^s(\theta^s(\alpha^a, \alpha^b))}.$$

$\forall \alpha^a, \alpha^b \in [0,1]$, increasing any $T_{yd}^s$ decreases $\frac{1 - g^{1s}(\theta^s(\alpha^a, \alpha^b))}{g^{0s}(\theta^s(\alpha^a, \alpha^b))}$. Let $\psi^{s'}(\cdot)$ be the consequent balanced function after increasing $T_{yd}^s$, and $\widehat{\alpha}^{s'}$ be corresponding equilibrium. Given any $\alpha^a, \alpha^b \in [0,1]$, we have $\psi^a(\alpha^b) < \psi^{a'}(\alpha^b)$ and $\psi^b(\alpha^a) < \psi^{b'}(\alpha^a)$. Therefore, $\widehat{\alpha}^{a'} > \widehat{\alpha}^a$ and $\widehat{\alpha}^{b'} > \widehat{\alpha}^b$. $\square$