[Reviews · NeurIPS 2020]

Review 1

Summary and Contributions: This paper proposes and analyzes a model of how fairness-oriented decision-making policies affect populations in the long run. The model takes into account feature distributions, qualification rates, and how they vary based on prior decisions. The authors focus their analysis on the equilibrium qualification rates. They derive theoretical results showing the effects of different fairness constraints on the evolution of these processes and support these results with simulated dynamics based on real-world datasets. Update after rebuttal: I appreciate the author feedback, and if accepted, I strongly recommend that they make the discussed changes to the COMPAS experiment.

Strengths: The model presented here is quite general and powerful, and I appreciate that the authors are able to derive concrete results in such a setting. The questions being studied here are quite relevant to the community, building upon a growing area of research. In this model, the authors do a good job of exploring and formalizing what were previously just intuitions about how the underlying state of the world modulates the impacts of various policies on long-term equity.

Weaknesses: This work is very closely related to a line of work in economics from the 90's onwards. I'd suggest starting with "Will affirmative-action policies eliminate negative stereotypes?" (Coate and Loury, 1993). Theorem 1 also seems related to some of the results in "Algorithmic Decision Making and the Cost of Fairness" (Corbett-Davies et al., 2017). The experiments section could use some improvement. In particular, the section on COMPAS data builds in the modeling assumption that "people who receive incarceration decisions are less likely to re-offend." First, this assertion needs justification. For example, "Gender differences in the effects of prison on recidivism" (Mears et al., 2012) and "Is Imprisonment Criminogenic?: A Comparative Study of Recidivism Rates between Prison and Suspended Prison Sanctions" (Cid, 2009) support the opposite conclusion. Second, COMPAS isn't used to make incarceration decisions; it's used to inform pre-trial detention decisions. Again, there's evidence that pre-trial detention increases recidivism risk (see, e.g., "The Hidden Costs of Pretrial Detention" (Lowenkamp et al., 2013)). I understand that the purpose of this experiment was to illustrate the effects of the model, but I don't think this is an appropriate use of this dataset. I'd encourage the authors to be a bit more careful and rigorous in their empirical analyses. The authors point out that interventions can be challenging due to "sensitivity to problem parameters." Is this a quirk of the model, or does this reflect a broader policy challenge? The paper's notation is quite dense. Given that the bulk of the paper focuses on a signle-dimensional feature space, it could be simplified to make it a bit easier to parse. I think it's worth mentioning somewhere in the paper the limitations of using group-specific policies. In many settings discussed by the authors (e.g., lending, criminal justice), such policies can be prohibited. Overall, I think this paper is quite interesting, and I believe it could be much stronger with some improvements.

Correctness: The claims and proofs appear to be correct, though I did not thoroughly check the appendix.

Clarity: As noted above, the notation can be tedious to parse. Other than that, the paper is fairly well-written.

Relation to Prior Work: See above.

Reproducibility: Yes

Additional Feedback: - line 33: "whether these static fairness can" -> "whether these static fairness constraints can"


Review 2

Summary and Contributions: This paper considers a repeated decisions framework, where each decision influences the underlying population. This builds on top of recently proposed frameworks on understanding the role of algorithmic fairness under such dynamics. The main extensions in the present framework are to allow indirect measurement of qualifications and group-dependent dynamics. This is done at the expense of some restrictions, such as strictly Markovian dynamics based on individual transitions, which are further restricted to have a unique equilibrium. The main contributions of the paper are to establish conditions for uniqueness, for fairness not to disturb natural equality at equilibrium, for fairness to improve inequality in the absence of natural equality. The paper also explores some tweaks to static fairness constraints that may improve the outlook in terms of equality or utility. Some (dynamics-) simulated experiments based on real data are also undertaken. (The supplementary material offer many additional insightful explorations too.)

Strengths: The dynamic framework of the paper adds certain desirable features that lacked in the prior literature of the kind, allowing for features of the qualifications instead of the qualifications directly to be observed, and allowing for differences in the dynamics between groups. The message in the results are reasonably crisp, and further the insight into this very hairy question. (Hairy namely because it is very difficult to present models that reasonably capture how society dynamically responds to decisions. Even in econometrics this is a big challenge. It is a fine balance between generality to capture all eventualities and specificity to derive meaningful conclusions.) The authors have clearly explored many ramifications of the clean formulation given in the main body of the paper. This is evident from the insights presented in the supplementary material.

Weaknesses: (I want to preface this section by saying that although I include a lot, it's mostly to round off the corners of the paper and not as much to argue against the paper, which I generally find strong.) Although true qualifications are latent and only features are observed, the paper assume a perfect knowledge of the Bayes optimal predictor of the qualifications. They do acknowledge that this is an idealization of a learned predictor. However, the real issue is that this assumption makes the framework only a minor extension of prior work: we have “soft” qualification measures instead of the “hard” measures assumed previously. Add to this the fact that the observed features are real-valued, and the threshold predictors being optimal follows immediately (from the mild monotonicity assumptions). Modeling choices are not always well-motivated. For example, why would the distribution of features given qualification depend on the group? I’m not saying this is unreasonable, but a justification is needed (e.g. cultural and perceptual factors). Similarly, focus on particular configurations is often justified in terms of the analytical complication rather than a natural restriction due to expected behavior (such as in the case of transition tendencies). While making the qualification transitions depend on the group is an extension, the fact that the transitions happen in response of individual decisions makes this strict Markovian dynamics, linear in the qualification rates which represent the state, Eq. (4). This does *not* capture phenomena such as an individual seeing the impact of the policy on its entire group and reacting accordingly: because this group-wide impact itself depends on the qualification rate of the group, that would break the linearity in Eq. (4). This is restrictive compared to some prior work. The focus on unique equilibria is somewhat restrictive. But what the authors fail to elaborate is the fact that even with the conditions that guarantee unique equilibria (Thm 3), there may still be oscillatory behavior due to the discrete steps of the dynamics. If I understand the proof correctly, the result seems to me more like local stability. I could be confused here, but regardless it may be worth clarifying whether limit cycles are possible or not, e.g. two points that repeatedly map to each other, because as it stands the theorem only claims uniqueness and doesn’t rule this out. In the experiments, we see this happen with the COMPAS model, despite the fact that the dynamics were simulated, I presume to the best possible conformance with the theory (though admittedly not all parameters can be controlled.) The weakest part of the paper is also one of its original selling points, and that’s new interventions that were suggested to deviate from the myopic policies to help in the long-term. (Prop 1 focuses only on absolute qualification, Prop 2 only on equality, and Prop 3 asks to modify something on which we have no control. Prop 1 and Prop 3 are somewhat unsurprising trade-offs, and Prop 2 does not say what is the price paid for equality with threshold policies, which can often be obtained by making everybody worse-off.)

Correctness: The paper is quite technical. I did check a couple of proofs, but only glanced over most of it. I did not find any errors in what I checked. (I was surprised why Thm 2 wasn’t proved using a fixed point theorem, until I saw the note in the end of the proof. It may be worth giving the alternative proof if it’s more transparent.) On Line 59 I can only assume it’s a typo to say that the features are in R^d. They’re only in R for all of the paper.

Clarity: The paper seems to assume that readers are familiar with the fairness under dynamics literature (namely references [36] and [44]). One thing that’s taken for granted is mixing the descriptions of the decision problem and the dynamics model. It’s worth delineating those for clarity. Upon establishing Thm 3, make it clear that the rest of the paper will assume uniqueness of the equilibrium.

Relation to Prior Work: Adequately covered, though more acknowledgement is necessary in the main paper (rather than only the supplements) to how much of the proposed framework was borrowed from [36] and [44].

Reproducibility: Yes

Additional Feedback: Assumption 1 is a reasonable monotonicity between x and y. Isn’t it better to phrase it as monotonicity of gamma? Assumption 2’s intuition is more muddy, and it may be worth explaining. It’s standard to say “equal opportunity” or “equalized opportunity” Consider renaming the “motivation” factor rather the “lack of motivation” factor. Another interesting question under natural equality is that if it’s not broken, is there any harm or benefit to the fairness constraints? Theorem 4 doesn’t quite address this. Explain why those particular scenarios (of features and transitions) were chosen under natural inequality In Thm 6, when DP flips the advantaged groups, do we know anything about the gap?


Review 3

Summary and Contributions: * The authors propose using partially-observed Markov decision processes (POMDPs) to study the long-term dynamics of decision-making. * They claim to prove that the optimal decision policy at each time-step is a threshold function over a single-dimensional value. * They characterize when the optimal threshold policy has a unique equilibrium in the long-term. * They analyze how fairness constraints affect the equilibrium, and find that under some conditions, fairness constraints produce a more equitable equilibrium and sometimes not. * They discuss the implications of their framework for suggested policy interventions. * They do simulations on completely synthetic data and synthetic data generated from two real-world datasets and find that the conclusions of their simulations align with typical policy recommendations.

Strengths: * Studying decision-making under dynamics is important, and it is good to see work on this topic. Because decision-making with dynamics is relatively new, there isn't a clear sense of the single best framework used to pursue these questions. Towards that end, the authors make a meaningful contribution by suggesting a particular formalism to tackle those questions (though I have reservations about the suitability of the formalism, as discussed below). * Sections 4 and 5 are insightful, within the assumptions of the POMDP framework. The authors provide an interesting way to characterize when fairness helps the long-term equilibrium by considering group differences in both initial feature distributions and the transitions between steps. For me, this insight is the distinguishing contribution of the paper. * The results are nuanced and do not provide a simplistic picture that says, "Fairness is good," or "Fairness is bad." * The authors make an effort to connect the results of their studies to the economics and social sciences literature. * The authors state their own beliefs about the limitations of this work.

Weaknesses: * The assertion that POMDPs are an appropriate framework for analyzing system dynamics requires further evidence and discussion. What situations would it be appropriate for, and when would it not be appropriate? What are the advantages/disadvantages of the POMDP framework? It's not a novel observation that most algorithmic decision are myopic because the true outcome (e.g. creditworthiness) are unobserved. And this same fact can be expressed equally well in other frameworks (e.g. a Bayesian one). * Furthermore, if the advantage of POMDPs is their assumption that the decision-maker does not directly observe the target variable Y, this advantage is not leveraged by the paper at all (for instance, it's not discussed how previous decisions changes the data observable to the decision-maker, since only outcomes are observed when the positive decision is given). * One disadvantage seems to be the Markov assumption: That the decision at any one point can only affect the hidden state Y and observed features X at the next time-step. It seems like many important decisions, such as loans or criminal risk assessment, actually use the entire history vector (consider that in making a loan decision, a bank might look to see the number of previous accepted/rejected loans)—which undermines the Markov assumption. * The author's choose a limited characterization of the different conditions of the system (the four possibilities stated within Condition 1), which involve comparing how agents react to decisions. These conditions all relate to how an individual will improve/fail to improve their status in the next round. For example, Condition 1(A) just states that a false positive hurt an individual's chances less than a true negative and true positive helps less than a false negative. The other three possible conditions (Conditions 1B-1D) are permutations of these relationships. This seems insufficient to characterize different dynamics; dynamics will also vary depending on the first-order effect of positive/negative classification on individuals' chances of success in the next round. In the language of the paper, it seems that one ought to also consider the relationship between avg(T_{01} + T_{11}) and avg(T_{00} + T_{10}). If the authors choose to note analyze of explore this dynamic in this paper, they could also rephrase their title and claims to make clear that their work focuses specifically on different individual response to decisions, rather than being a general paper about "fair decisions and long-term qualifications." * The authors claim that intervening upon transitions is a useful way to improve the dynamics of systems. Particularly, one should improve the transitions T_{01} that indicate the change of behavior of individuals who are qualified but are nonetheless rejected by the policy. While this is a neat idea, in practice this will be difficult to achieve, since "true' outcomes are not observed for people who do not receive a certain decision and so the subpopulation of individuals who follow T_{01} is unknown. * One of the main results of the paper is overstated. The authors claim to prove that "the optimal myopic policy is a threshold policy" (lines 44-45). This is reinventing the wheel, where the wheel is standard binary classification (without dynamics). That a threshold policy is still optimal for binary classification with fairness constraints is a newer result, but can be found in the work of Corbett-Davies et al. (2017): http://arxiv.org/abs/1701.08230. This result is probably better stated as an assumption or previous result that is used throughout the rest of the paper.

Correctness: * In the weaknesses section, I have detailed a number of conceptual concerns with the methodology. These concerns don't take away from the correctness of the specific results in the paper, but rather, question the theoretical assumptions made by the framework and its applicability to real-world scenarios. * The empirical methodology relies on a number of assumptions, which is inevitable since the authors do not (nor does anybody else, to my knowledge) have access to datasets that capture the effect decision-making over time. While in general these assumptions are OK, I have to call out one important oversight: In lines 345-347, the authors claim that incarceration are less likely to re-offend in the future, and run their simulations accordingly. It's actually not at all clear that incarceration reduces recidivism (one negative account is: https://journals.sagepub.com/doi/abs/10.1177/0032885511415224; though not definitive, it's worth considering, since these authors state that incarceration may in fact increase recidivism!), and it would have been better if the authors had run their COMPAS simulations to account for this uncertainty. As a friendly word of advice in this politicized environment, it might be good for the authors to acknowledge this uncertainty and model it in their work (by running the COMPAS simulations with different conditions on the transitions). But this oversight does not undermine the result the authors are trying to show, which is that there are different equilibrium depending on the underlying dynamics of the system. * I did not examine the proofs in the appendix fully, but I think it's plausible that these results are correct. Finding equilibrium in dynamical systems has received much study, and it doesn't seem to me that the authors have proved something that is extraordinarily unintuitive which would require an extraordinary proof.

Clarity: * Generally, there is a sense of coherence throughout the entire paper. The main claims are stated at the beginning, and the results are presented in a logical order. However, there is work to be done within individuals sections: * Some technical sections require more exposition. There are many new symbols to keep track of, as the authors favor introducing many new functions to simplify the appearance of complicated expressions. It would be better to include some motivation or intuition for what these results mean (e.g. explaining the sufficient conditions for equilibrium, as stated in Theorem 3). * Relatedly, many theorems are stated as facts without any exposition in the main paper. It's OK that the details can be left to an appendix, but it would be more convincing if the authors could include a proof sketch in the primary article. * Condition 2 is difficult to follow, as it is not clear when each clause of the conjunction begins and ends. * Line 12: extra "the" * Line 159: Notation is unclear. Perhaps the authors mean to say, "highly depends on the transitions T for all possible values of y and d"?

Relation to Prior Work: The authors discuss related work in the appendix and state clearly how their work differs from others (not assuming that decision-maker has access to the ground truth, which is a realistic and good assumption to work from).

Reproducibility: Yes

Additional Feedback: Just to explain my score, I would like to have given a 6.5, if possible. My rating of a 6 follows the additional guideline in the Reviewers Guide that a 6 corresponds to, "I tend to vote for accepting this submission, but rejecting it would not be that bad." The authors have chosen an interesting and important topic area and, within the confines of their framework, done their technical due diligence, but I think the conceptual thinking in this paper is a bit weak. Update after author rebuttal: Overall, I feel confident that the authors will take the feedback we've given them and use it to improve the paper. Though I don't feel completely as if my feedback has been completely resolved, there may just be some conceptual disagreements between the authors and myself that we won't be able to work out during the review process. But it's probably a good thing to have these debates in public, so I would also go for acceptance in that case. I've changed my score from a 6 to a 7 accordingly. R2 can speak for themself on their other reasons for marking the paper slightly below the threshold, but I do want to highlight that I think it would really important to make sure that the authors address the COMPAS experiment in their revisions. I would be happy with the authors' suggestion of showing results for all possible conditions in the main paper with a more nuanced discussion about the uncertainty around the impact of incarceration.

[Author Response · NeurIPS 2020]

R2 & R5: "Threshold policies are optimal". We thank **R2**, **R5** for the reference (Corbett-Davies et al., 2017). Our Thm
1 is indeed similar. We will be sure to cite this work and explain the relationship with ours.

R2 & R5: "COMPAS experiments". We share the same
reservation in using this dataset and lending validity to Con-
dition 1A, even though our purpose was only to show the
flexibility of our framework. As an alternative, we've run
experiments on COMPAS under all conditions (1A-D). Table
1 shows Prop 1 holds under 1A-B, no oscillation under 1B-
C, and more uncertainty under 1C-D, which is discussed in
Appendix F. We will replace the original version with these
results, or, if this still does not address reviewers' concerns,
we are fine with removing this experiment altogether.

Table 1: $\text{osi}^*/\text{osi}_H/\text{osi}_L$ is the percentage that oscillation occurs among 125 set of different transitions under policy $\text{UN}^*/\text{UN}_{\theta_H}/\text{UN}_{\theta_L}$. Among transitions that lead to stable equilibrium, Col 2/Col 3 shows the percentage that $\text{UN}_{\theta_H}/\text{UN}_{\theta_L}$ results in lower recidivism compared with $\text{UN}^*$.

| | $\widehat{\alpha}_{\theta_H} < \widehat{\alpha}^*$ | $\widehat{\alpha}_{\theta_L} < \widehat{\alpha}^*$ | $\text{osi}^*$ | $\text{osi}_H$ | $\text{osi}_L$ |
|---|---|---|---|---|---|
| $A$ | 0 | 1 | 0.29 | 0.12 | 0.36 |
| $B$ | 0.99 | 0.01 | 0 | 0 | 0 |
| $C$ | 0.37 | 0.28 | 0 | 0 | 0 |
| $D$ | 0.79 | 0.63 | 0.06 | 0 | 0.13 |

R2 & R3: "1D feature space". We thank **R2** and **R3** for suggestions on notations. To clarify, our work is not limited to
1D feature space (line 115-116); the generalization to $\mathbb{R}^d$ is in Appendix E.

R2, R3 & R5: "Interventions". **@R2** sensitivity: This is one of our main findings (lines 7-9) and reflects broader
challenges in designing a fair policy: the effect of intervention highly depends on problem parameters, and the same
intervention may lead to contrarian results as parameters change. **@R3** Trade-off: Prop. 1 improves $\widehat{\alpha}^s$ by sacrificing
instant utility; Prop. 2 achieves equality but violates static fairness and sacrifices instant utility. Note that the sacrifice
in instant utility in both cases may actually result in improved long-term total utility (line 283). While $T^s_{yd}$ is a group
property, it is controllable through community-level interventions (Prop. 3) such as social support (subsidy, training, etc.)
to sub-populations (line 294). **@R5** usage: Intervention remains feasible even without knowing the true qualification
(of the rejected population); e.g., supporting all those rejected (or accepted) would increase both $T_{1d}$ and $T_{0d}$.

R2: "Related work in economics". The model studied in (Coate and Loury, 1993) is more relevant to [33], where people
manipulate their qualifications and groups have identical feature distribution and response to the policy. We will add
this comparison and more related works in economics. "Limitations on group-specific policies". In some cases the use
of sensitive attribute is allowed (e.g., per ECOA Regulation B, *age* can be used in lending in the US). Nonetheless, we
will clarify that group-specific policies may not be generally applicable.

R3:"Stability of unique equilibrium". It is true there can be oscillation under conditions in Thm.3, as they only guarantee
uniqueness but not stability. We discuss stability in Appendix F (line 662-677), and will clarify this in the main body.
"Harm/benefit of fairness if natural equality is not broken". This is examined in Thm.4 (line 218-219): equality is
violated/maintained if distributions are different/same."Scenarios under natural inequality".Under our model two groups
can be different in transition or/and feature distribution. Natural inequality arises when either one or both differ across
two groups. We thus regard these as two sources of inequality, consider them separately by fixing one and varying the
other (two scenarios we studied), and examine whether fairness constraints can address inequality caused by each (line
228-230). "When DP flips the advantaged group". In this case the gap $|\widehat{\alpha}^a_{\text{DP}} - \widehat{\alpha}^b_{\text{DP}}|$ highly depends on feature distribution.
Empirical results (Table 3, Appendix A) show DP can reduce this gap (mitigate inequality): $\widehat{\alpha}^b_{\text{DP}} - \widehat{\alpha}^a_{\text{DP}} < \widehat{\alpha}^a_{\text{UN}} - \widehat{\alpha}^b_{\text{UN}}$.

R3 & R5: "Markov dynamics". As long as an appropriate "state" (sufficient statistics) can be identified, the Markov
assumption holds; this in the worst case would be the entire history which would indeed be undesirable. In practice
historical information is often summarized into a (pseudo) sufficient statistics to enable tractable decision making; e.g.,
lending decisions rely on the entire history only through summaries such as the credit score or a set of scores, which
can be regarded as the state in a Markov process. **@R3** linearity: Although individual action doesn't depend on $\alpha^s_t$, we
note Eqn. (4) is *not* linear in qualification $\alpha^s_t$, as $g^{1s}_t$ and $g^{0s}_t$ are functions of policy $\pi^s_t$, which is nonlinear in $\alpha^s_t$.

R3 & R5: "Justification on modeling choices". **@R3**: In addition to current explanations and examples (line 162-165 on
transitions; line 234-236, 254-255 on two scenarios under natural inequality), we will further strengthen the motivation
for these modeling choices. **@R5** POMDP: We think POMDP is a reasonable framework to capture the sequential
nature of the problem and the fact the true qualifications $Y$ are unobservable to the decision maker, and the decision is
based on $X$ and a belief state on $Y$. A (PO)MDP approach has been motivated and used in similar studies [9,23]. In
particular, [9] shows that although many works on fairness didn't explicitly use (PO)MDP to model dynamics, they
can all be cast into the standard framework of (PO)MDP, such as works on lending [32,36], college admission [22,28],
attention allocation [11], etc. The studies on the scenario mentioned by **R5** (outcomes are observed only under positive
decisions) are orthogonal to our work, which we introduce and discuss in Appendix C (line 596-598).

R5: "Conditions 1A-D". Under our POMDP framework (Fig. 1), dynamics of $\alpha_t$ follow Eqn. (4) and 1A-1D actually
capture all possibilities including the case mentioned by the reviewer. Specifically, if $\text{avg}(T_{01} + T_{11}) \leq$ (resp. $\geq$)
$\text{avg}(T_{00} + T_{10})$ holds, then either 1A (resp. 1B) or 1C or 1D must hold. "Explain conditions in Thm. 3". This can be
considered as a weaker version of the Lipschitz condition. More discussion is in Appendix F (line 662-677).

[Meta-Review · NeurIPS 2020]

The paper investigates the problem of fairness when decisions affect future distributions, using a POMDP model. Three knowledgeable reviewers have carefully considered the strengths and weaknesses of the paper, with two reviewers recommending accept, and one just below the threshold. The author rebuttal was considered, and the reviewers discussed their ratings, as well as adjusted some aspects of their reviews. Reviewer #3 raised many issues, but clearly stated that the paper is strong, and was positive during the post rebuttal discussions. Reviewer #2 and #5 had concerns about the conceptual framing of the problem, and which I feel is balanced by the novelty of the setting considered. All reviewers identified improvements that can be made on the empirical results. During discussion, the reviewers agreed that the setting is interesting and a timely topic, and while there are rough edges to the execution, I recommend to accept for NeurIPS 2020. I strongly encourage the authors to update the paper to address all the valid issues observed by the reviewers for the final version.